# ELRT: Efficient Low-Rank Training for Compact Convolutional Neural Networks

## Abstract

Low-rank compression, a popular model compression technique that produces compact convolutional neural networks (CNNs) with low rankness, has been well studied in the literature. On the other hand, low-rank training, as an alternative way to train low-rank CNNs from scratch, is little exploited yet. Unlike low-rank compression, low-rank training does not need pre-trained full-rank models and the entire training phase is always performed on the low-rank structure, bringing attractive benefits for practical applications. However, the existing low-rank training solutions are still facing several challenges, such as considerable accuracy drop and/or still needing to update full-size models during the training. In this paper, we perform a systematic investigation on low-rank CNN training. By identifying the proper low-rank format and performance-improving strategy, we propose ELRT, an efficient low-rank training solution for high-accuracy high-compactness low-rank CNN models. Our extensive evaluation results for training various CNNs on different datasets demonstrate the effectiveness of ELRT.

## 1 Introduction

Convolutional neural networks (CNNs) have obtained widespread adoption in numerous real-world computer vision applications, such as image classification, video recognition and object detection. However, modern CNN models are typically storage-intensive and computation-intensive, potentially hindering their efficient deployment in many resource-constrained scenarios, especially at the edge and embedded computing platforms. To address this challenge, many prior efforts have been proposed and conducted to produce low-cost compact CNN models. Among them, **low-rank compression** is a popular model compression solution. By leveraging matrix or tensor decomposition techniques, low-rank compression aims to explore the potential low-rankness exhibited in the full-rank CNN models, enabling simultaneous reductions in both memory footprint and computational cost. To date, numerous low-rank CNN compression solutions have been reported in the literature (Phan et al. (2020); Kossaifi et al. (2020); Li et al. (2021b); Liebenwein et al. (2021)).

**Low-rank Training: A Promising Alternative Towards Low-rank CNNs.** From the perspective of model production, performing low-rank compression on the full-rank networks is not the only approach to obtaining low-rank CNNs. In principle, we can also adopt low-rank training strategy to *directly train a low-rank model from scratch*. As illustrated in Fig. 1, low-rank training starts from a low-rank initialization and keeps the desired low-rank structure in the entire training phase. Compared with low-rank compression that is built on two-stage pipeline ("pre-training-then-compressing"), the single-stage low-rank training enjoys **two attractive benefits:** *relaxed operational requirement* and *reduced training cost*. More specifically, first, the underlying training-from-scratch scheme, by its nature, completely eliminates the need for pre-trained full-rank high-accuracy models, thereby lowering the barrier to obtaining low-rank CNNs. In other words, producing low-rank networks becomes more feasible and accessible. Second, the overall computational cost for the entire low-rank CNN production pipeline is significantly reduced. This is because: 1) the removal of the pre-training phase completely saves the incurred computations that were originally needed for pre-training the full-rank models; and 2) directly training on the compact low-rank CNNs naturally consumes much fewer floating point operations (FLOPs) than full-rank pre-training.

**Existing Works and Limitations.** Despite the above analyzed benefits, low-rank training is currently little exploited in the literature. Unlike the prosperity of studying low-rank compression, to

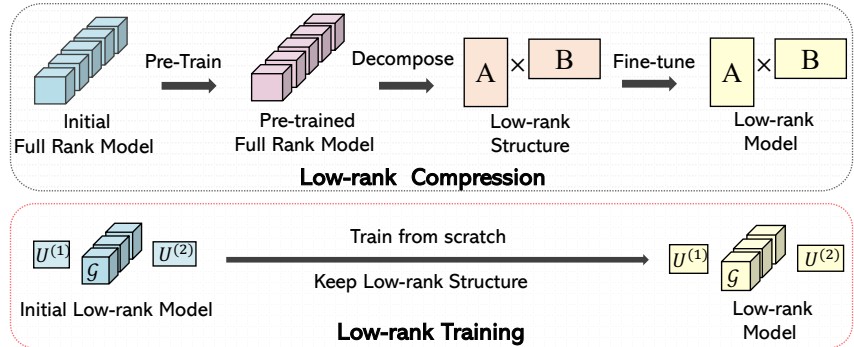

Figure 1: Different paths towards producing low-rank CNN models.

date very few research efforts have been conducted towards efficient low-rank training. (Ioannou et al. (2015); Tai et al. (2015)) are the pioneering works in this research direction; however the obtained low-rank models suffer considerable accuracy drop. In addition, the corresponding training methods are not evaluated on the modern CNNs such as ResNet. Recently, (Gural et al. (2020); Hawkins et al. (2022)) propose emerging memory-aware and Bayesian estimation-based low-rank training, respectively; however, these two method are either built on costly repeated SVD operations (Gural et al. (2020)) or Bayesian estimation (Hawkins et al. (2022)), which are very computation intensive and potentially not scalable for practical deployment. (Hayashi et al. (2019); Khodak et al. (2020); Su et al. (2022)) propose to learn the suitable low-rank format and/or apply spectral initialization during the training. However, the trained models with the new format have a considerable accuracy drop, even on CIFAR-10 dataset. (Waleffe & Rekatsinas (2020); Wang et al. (2021)) perform several epochs of full-size training to mitigate this issue. However, with the cost of increasing memory and computational overhead, this hybrid-training strategy still brings a considerable accuracy drop. And it is essentially not training low-rank model from scratch. Therefore, the satisfactory answer to the following fundamental question is still missing:

**Fundamental Question for Low-rank Training:** *What is the proper training-from-scratch solution that can produce modern low-rank CNN models with high accuracy on the large-scale dataset, even outperforming the state-of-the-art low-rank compression methods?*

**Technical Preview and Contributions.** To answer this question and put the low-rank training technique into practice, in this paper, we perform a systematic investigation on training low-rank CNN models from scratch. By identifying the proper low-rank format and performance-improving strategy, we propose ELRT, an efficient low-rank training solution for high-accuracy high-compactness CNN models. Compared with the state-of-the-art low-rank compression approaches, ELRT demonstrates superior performance for various CNNs on different datasets, demonstrating the promising potential of low-rank training in practical applications. Overall, the contributions of this paper are summarized as follows:

- We systematically investigate the important design knobs of low-rank CNN training from scratch, such as the suitable low-rank format and the potential performance-improving strategy, to understand the key factors for building a high-performance low-rank CNN training framework.

- Based on the study and analysis of these design knobs, we develop ELRT, an orthogonality-aware low-rank training approach that can train high-accuracy high-compactness low-tensor-rank CNN models from scratch. By enforcing and imposing the desired orthogonality on the low-rank model during the training process, significant performance improvement with low computational overhead can be obtained.

- We conduct empirical evaluations for various CNN models to demonstrate the effectiveness of ELRT. On CIFAR-10 dataset, ELRT can train low-rank ResNet-20, ResNet-56 and MobileNetV2 from scratch with providing $1.98\times$, $2.05\times$ and $1.71\times$ FLOPs reduction, respectively; and meanwhile the trained compact models enjoy $0.48\%$, $0.70\%$ and $0.29\%$ accuracy increase over the baseline. On ImageNet dataset, compared with the state-of-the-art approaches that generate compact ResNet-50 models, ELRT achieves $0.49\%$ higher

accuracy with the same or even higher inference and training FLOPs reduction, respectively.

## 2 RELATED WORKS

**Low-rank Compression.** As an important type of model compression strategy, low-rank compression aims to leverage low-rank decomposition techniques to factorize the original full-rank neural network model into a set of small matrices or tensors, leading to storage and computational savings. Based on the adopted factorization methods, the existing low-rank CNN compression works can be categorized into *2-D matrix decomposition based* (Tai et al. (2015); Li & Shi (2018); Xu et al. (2020); Idelbayev & Carreira-Perpinán (2020); Yang et al. (2020); Liebenwein et al. (2021)) and *high-order tensor decomposition based* (Denton et al. (2014); Kim et al. (2015); Novikov et al. (2015); Yang et al. (2017); Wang et al. (2018); Kossaifi et al. (2019); Phan et al. (2020); Kossaifi et al. (2020); Li et al. (2021a) Lin et al. (2020b); Yu et al. (2021)).

**Low-rank Training.** Similar to low-rank compression, the goal of low-rank training is also to produce compact neural network models with low-rankness; while the key difference is that low-rank training initializes and updates the low-rank CNNs during the entire training process. In other words, the pre-trained full-rank models are not required in this scenario, and the CNN models being updated are always kept in the low-rank format. To date efficient low-rank training approaches are still little exploited. More specifically, the existing works either have considerable accuracy loss (Ioannou et al. (2015); Tai et al. (2015) Hayashi et al. (2019); Khodak et al. (2020); Su et al. (2022)) or suffer high computational overhead because of the use of costly SVD operations (Gural et al. (2020)), Bayesian estimation (Hawkins et al. (2022)) or only performing partially low-rank training (Waleffe & Rekatsinas (2020); Wang et al. (2021)), limiting their effectiveness in the practical scenarios.

**Unstructured & Structured Sparse Training.** Low-rank training is essentially a type of *compression-aware training* solutions, which include another related strategy as sparse training. Sparse training can be performed in the unstructured (Lee et al. (2018); Wang et al. (2019); Evci et al. (2020); Mostafa & Wang (2019); Liu et al. (2020); Mocanu et al. (2018); Bellec et al. (2018)) and structured (Yuan et al. (2021); Zhou et al. (2021)) ways, corresponding to training/obtaining unstructured and structured sparse CNN models, respectively. From the perspective of model deployment, structured sparse training is more practical and important than its unstructured counterpart because it can produce structured sparse CNN models that exhibit considerable speedup on the off-the-shelf GPU/CPU platforms.

**Orthogonality in CNN Training.** Another line of works that are related to this paper is training orthogonal full-size full-rank CNNs (Rodriguez et al. (2017); Huang et al. (2018); Miyato et al. (2018); Bansal et al. (2018); Wang et al. (2020)). Based on the observation that orthogonality in CNN weights can stabilize the distribution of activations and bring efficient optimizations, these prior efforts explore different methods to enforce orthogonality on convolutional layer in both initialization and training phases. Compared with them, **ELRT has two key differences.** First, the existing orthogonal CNN training works are performed on full-rank full-size models, and they cannot bring any memory or FLOPs reduction. Instead, ELRT aims to train compact low-rank CNN models from scratch, naturally bringing reduced inference and training costs. Second, orthogonal CNN training directly enforces orthogonality on the entire convolutional layer. The underlying motivation is mainly based on experimental observation (e.g., stabilizing the distribution of activations). On the other hand, ELRT focuses on exploring the orthogonality of the decomposed components (e.g., factor matrices) of the weight tensors. Such strategy is naturally consistent with the requirement of low-rank theory – the decomposed matrix/tensor should exhibit self-orthogonality after SVD or tensor decomposition.

## 3 PRELIMINARIES

**Notation.** Throughout this paper the $d$-order tensor, matrix and vector are represented by boldface calligraphic script letter $\boldsymbol{\mathcal{X}} \in \mathbb{R}^{n_1 \times n_2 \times \cdots \times n_d}$, boldface capital letters $\boldsymbol{X} \in \mathbb{R}^{n_1 \times n_2}$, and boldface lower-case letters $\boldsymbol{x} \in \mathbb{R}^{n_1}$, respectively. Also, $\boldsymbol{\mathcal{X}}_{i_1, \cdots, i_d}$ and $\boldsymbol{X}_{i,j}$ denote the entry of tensor $\boldsymbol{\mathcal{X}}$ and matrix $\boldsymbol{X}$, respectively.

**Tucker-2 Format for Convolutional Layer.** As will be analyzed in Section 4, in this paper we choose to use low-tensor-rank format (e.g., Tucker-2) to form efficient low-rank training approach. In general, given a convolutional layer $\mathcal{W} \in \mathbb{R}^{C_{\text{in}} \times C_{\text{out}} \times K \times K}$, it can be represented in a Tucker-2 format as follows:

$$\mathcal{W}_{p,q,i,j} = \sum_{r_1=1}^{\Phi_1} \sum_{r_2=1}^{\Phi_2} \mathcal{G}_{r_1,r_2,i,j} \mathbf{U}_{r_1,p}^{(1)} \mathbf{U}_{r_2,q}^{(2)}, \tag{1}$$

where $\mathcal{G}$ is the 4-D **core tensor** of size $\Phi_1 \times \Phi_2 \times K \times K$, and $\mathbf{U}^{(1)} \in \mathbb{R}^{\Phi_1 \times C_{\text{in}}}$ and $\mathbf{U}^{(2)} \in \mathbb{R}^{\Phi_2 \times C_{\text{out}}}$ are the **factor matrices**. In addition, $\Phi_1$ and $\Phi_2$ are the tensor ranks that determine the complexity of compact convolutional layer.

**Computation on the Tucker-2 Convolutional Layer.** Given the above described Tucker-2 format representation, the corresponding execution on this compact convolutional layer can be performed via consecutive computations as:

$$\mathcal{T}_{r_1,h,w}^1 = \sum_{p=1}^{C_{\text{in}}} \mathbf{U}_{r_1,p}^{(1)} \mathcal{X}_{p,h,w}, \quad \mathcal{T}_{r_2,h',w'}^2 = \sum_{i=1}^{K} \sum_{j=1}^{K} \sum_{r_1=1}^{\Phi_1} \mathcal{G}_{r_1,r_2,i,j} \mathcal{T}_{r_1,h_i,w_j}^1,$$

$$\mathcal{Y}_{q,h',w'} = \sum_{r_2=1}^{\Phi_2} \mathbf{U}_{r_2,q}^{(2)} \mathcal{T}_{r_2,h',w'}^2, \tag{2}$$

where $\mathcal{X} \in \mathbb{R}^{C_{\text{in}} \times H \times W}$ and $\mathcal{Y} \in \mathbb{R}^{C_{\text{out}} \times H' \times W'}$ are the input and output tensor of the convolutional layer, respectively. For indices of $\mathcal{T}^1$, $h_i = \texttt{stride} \times (h'-1) + i - \texttt{padding}, w_j = \texttt{stride} \times (w'-1) + j - \texttt{padding}$. In addition, $\mathcal{T}^1$ and $\mathcal{T}^2$ are the incurred intermediate results.

## 4 PROPOSED LOW-RANK TRAINING SOLUTION

As outlined in Section 1, to date the efficient training for high-accuracy low-rank CNN models from scratch is still largely under-explored. In this section we propose to systematically explore several important design knobs and factors when training low-rank CNN models from scratch. Based on the outcomes from these analytic and empirical studies, we will then further develop efficient solution for low-rank CNN training from scratch.

**Questions to be Answered.** To be specific, in order to obtain better understanding of low-rank training and improve its performance, we explore the answers to the following three questions.

**Question1:** *Which type of low-rank format is more suitable for efficient training from scratch, 2-D matrix or high-order tensor?*

Analysis. In general, when training a compact CNN model from scratch, there are two types of low-rank formats that can be considered. The 4-D weight tensor of the trained convolutional layer can be either in the format of a low-matrix-rank 2-D matrix or a high-order low-tensor-rank tensor. As illustrated in Fig. 2, the low-matrix-rankness means the flattened and matricized 4-D weight tensor exhibits low-rankness; while the low-tensor-rankness means the trained convolutional layer can be directly represented and constructed via multiple small-size factorized matrices/tensors without any flattening operations.

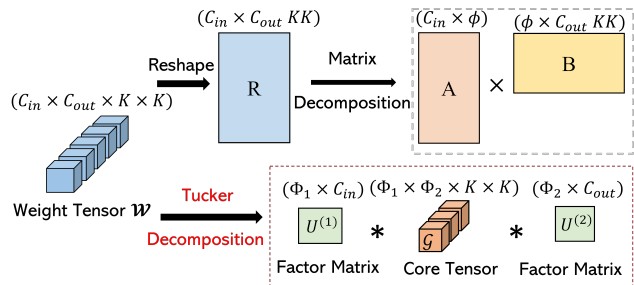

Figure 2: A low-rank CONV layer can either exhibit low-matrix-rankness (Top) or low-tensor-rankness (Bottom).

Our Proposal. Currently most of the existing low-rank training works (Yang et al. (2020); Ioannou et al. (2015); Tai et al. (2015)) conduct and keep 4-D convolutional layer training in the format of low-rank 2-D matrix. Instead, we propose to perform low-rank CNN training directly

in the high-order tensor format. In other words, each convolutional layer always stays in the low-rank tensor decomposition format, e.g., Tucker (Tucker (1966)) or CP (Hitchcock (1927)), during the entire training phase. Our rationale for this proposal is that, unlike low-rank matrix format that may lose the important spatial weight correlation incurred by the inevitable flattening operation; the low-rank tensor format is a more natural way to represent 4-D weight tensor of convolutional layer; and therefore it can better extract and preserve the weight information and correlation existed in the 4-D space (Liu et al. (2012)). For instance, as illustrated in Fig. 3, when representing the same weight tensors of layers in ResNet-20 model, the low-rank Tucker format enjoys a smaller approximation error than the low-rank matrix format with the same number of weight parameters. Encouraged by this better representation

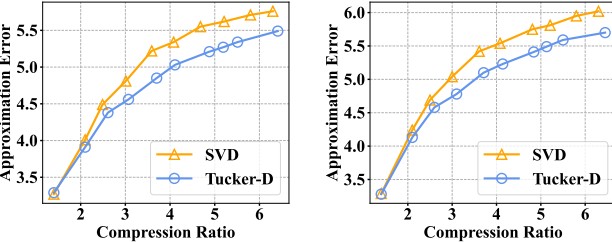

(a) The 15-th convolutional layer.

(b) The 16-th convolutional layer.

Figure 3: Approximation error (Mean Square Error (MSE)) of low-matrix-rank and low-tensor-rank methods for approximating ResNet-20 layers. Notice that MSE measurement is our analysis and exploration to identify the suitable low-rank format. It is not actually executed during training.

capability for the high-order weight tensors, we choose to train low-rank tensor format CNN from scratch.

**Question2:** *Consider low-rankness implies potentially low model capacity, what is proper strategy to improve the performance of low-rank training?*

Analysis. As analyzed in Section 1, a key challenge of the existing low-rank training works (Tai et al. (2015); Ioannou et al. (2015)) is their inferior model accuracy as compared to model compression approaches. We hypothesize that this phenomenon is caused by two reasons: 1) low-rank training starts from a low-accuracy random initialization with low-rank constraints; while model compression is built on a pre-trained high-accuracy model; and 2) the entire procedure of low-rank training is constrained in the low-rank space, thereby limiting the growing space for model capability. Notice that a possible solution is that we can reconstruct and train a full-rank model in some epochs, and then decompose the model to low-rank format again during the training procedure. However, such "train-decompose-train" scheme is very costly since it needs to perform computation-intensive tensor decomposition many times.

Our Proposal. To improve the performance of low-rank training with preserving low computational and memory costs, we propose to perform orthogonality-aware low-tensor-rank training. Our key idea is to impose and enforce the orthogonality on the factor matrices $\mathbf{U}^{(1)}$ and $\mathbf{U}^{(2)}$ during the entire training process, and such training philosophy lies in the following rationale: It is well known that in principle using orthogonal-format basis can maximize the capacity of information representation. For instance, when we aim to approximate a full-rank matrix with low-rank SVD factorization, the decomposed $\mathbf{U}$ and $\mathbf{V}$ always exhibit self-orthogonality (as unitary matrix) with the smallest approximation error. A similar phenomenon also exists for tensor decomposition, where the factor matrices $\mathbf{U}^{(1)}$ and $\mathbf{U}^{(2)}$ are also unitary matrices after performing Tucker decomposition on the original full-rank tensor. Inspired by above observations, when we aim to approach a high-capacity full-rank model using the low-rank format, the orthogonality-based representation is

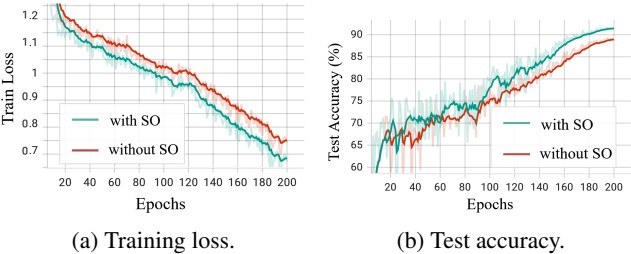

(a) Training loss.

(b) Test accuracy.

Figure 4: Training loss (left) and test accuracy (right) for low-tensor-rank ResNet-20 on CIFAR-10 with/without SO regularization. Same ranks are used for different experiments. Ranks are selected to provide 2× FLOPs reduction.

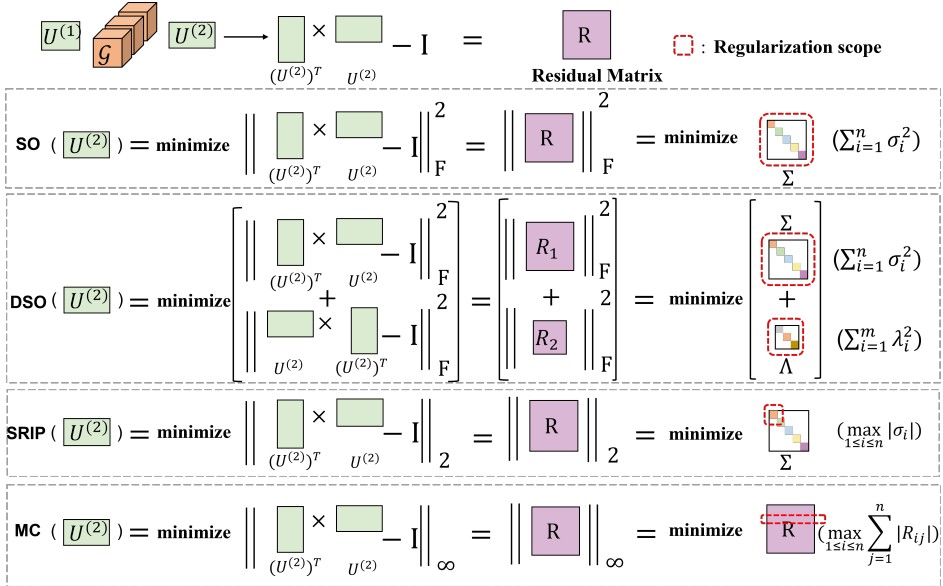

Figure 5: The mechanism of different approaches to impose orthogonality on $\mathbf{U}^{(2)}$. From top to bottom: (a) Soft Orthogonal Regularization, (b) Double Soft Orthogonal Regularization, (c) Spectral Restricted Isometry Property Regularization, (d) Mutual Coherence Regularization.

a very suitable solution. However, such orthogonality cannot be automatically obtained and ensured during the low-rank training. Therefore, an efficient mechanism of enforcing and ensuring orthogonality is very desired.

To verify the potential effectiveness of the proposed orthogonality-aware idea, we perform a preliminary ablation study of low-tensor-rank training with and without enforcing orthogonality. Here a ResNet-20 model is trained from scratch on CIFAR-10 dataset, and a simple soft orthogonal regularization term (Xie et al. (2017); Bansal et al. (2018)) is used to impose orthogonality on $\mathbf{U}^{(1)}$ and $\mathbf{U}^{(2)}$ during the training. As shown in Fig. 4, enforcing orthogonality during the training can significantly improve training and testing performance.

**Question3:** *How should we properly impose the orthogonality during the low-tensor-rank training?*

Analysis. The preliminary experiment in Fig. 4 shows the encouraging benefits of orthogonality-aware low-tensor-rank training. To fully unlock its promising potentials, efficient scheme for imposing the desired orthogonality should be properly explored. To be specific, there are four commonly used regularization approaches (Bansal et al. (2018)) that can introduce the orthogonality on the target factor matrices:

*Soft Orthogonal (SO) Regularization:*

$$\mathcal{R}_s(\mathbf{A}) = \frac{\rho}{\Phi^2}\|\mathbf{A}^{\mathrm{T}}\mathbf{A} - \mathbf{I}\|_F^2, \tag{3}$$

where $\|\cdot\|_F$ is the Frobenius norm, $\rho$ is regularization strength, $\mathbf{A}$ is the matrix to be enforced with orthogonality, and $\Phi$ is the rank of $\mathbf{A}$. Notice the experiment in Fig. 4 adopts this approach.

*Double Soft Orthogonal (DSO) Regularization:*

$$\mathcal{R}_d(\mathbf{A}) = \frac{\rho}{\Phi^2}(\|\mathbf{A}^{\mathrm{T}}\mathbf{A} - \mathbf{I}\|_F^2 + \|\mathbf{A}\mathbf{A}^{\mathrm{T}} - \mathbf{I}\|_F^2). \tag{4}$$

Here compared with SO regularization, this DSO scheme can always impose the proper orthogonality on $\mathbf{A}$ no matter it is over-complete or under-complete.

*Mutual Coherence (MC) Regularization:*

$$\mathcal{R}_{mc}(\mathbf{A}) = \rho(\|\mathbf{A}^{\mathrm{T}}\mathbf{A} - \mathbf{I}\|_{\infty}), \tag{5}$$

where $\| \cdot \|_\infty$ is the matrix norm induced by the $\ell_\infty$-norm.

*Spectral Restricted Isometry Property (SRIP) Regularization:*

$$\mathcal{R}_{sp}(\mathbf{A}) = \rho \cdot \sigma(\mathbf{A}^{\mathrm{T}}\mathbf{A} - \mathbf{I}), \tag{6}$$

where $\sigma(\mathbf{R}) = \sup_{z \in \mathbf{R}^n, z \neq 0} \frac{\|\mathbf{R}z\|^2}{\|z\|^2}$ is the spectral norm of $\mathbf{A}^{\mathrm{T}}\mathbf{A} - \mathbf{I}$ (Yoshida & Miyato (2017)) based on RIP condition (Candes & Tao (2005)).

Our Proposal. We propose to develop DSO regularized scheme to impose the desired orthogonality on the factor matrices. Here our main rationale is that DSO regularization can better provide the desired orthogonality. To be specific, as illustrated in Fig. 5, the orthogonality of the factor matrix $\mathbf{U}^{(2)}$ can be measured by its corresponding residual matrix $\mathbf{R} = \mathbf{U}^{(2)^{\mathrm{T}}}\mathbf{U}^{(2)} - \mathbf{I}$. Here when $\mathbf{R}$ has lower energy, it is more likely that $\mathbf{U}^{(2)}$ will exhibit more self-orthogonality. Therefore, the essential mechanism of SRIP and MC is to minimize the maximum singular value of $\mathbf{R}$ and the maximum row sum norm of $\mathbf{R}$, respectively, to push $\mathbf{R}$ close to all-zero matrix. Evidently, such constraint posed by SRIP and MC schemes inherently targets to the local components of $\mathbf{R}$, thereby degrading the overall effect of orthogonality. On the other hand, DSO and SO schemes aim to push all the singular values of $\mathbf{R}$ to zero in a global way, thereby in principle bringing stronger orthogonality for $\mathbf{U}^{(2)}$. In addition, since $\mathbf{U}^{(1)}$ and $\mathbf{U}^{(2)}$ may not be square matrix in practice, DSO scheme, as an approach that simultaneously considers the potential under-completeness and over-completeness of matrix, is a more general and better solution than SO scheme.

---

**Algorithm 1:** Overall ELRT training procedure

**Input:** Dataset $\mathcal{D}$, pre-set ranks $\{\Phi\}$, Tucker-2-format weights in each layer $\mathbf{U}^{(1)} \in \mathbb{R}^{\Phi_1 \times C_{in}}, \boldsymbol{\mathcal{G}} \in \mathbb{R}^{\Phi_1 \times \Phi_2 \times K \times K}, \mathbf{U}^{(2)} \in \mathbb{R}^{\Phi_2 \times C_{out}}$, orthogonal parameters $\rho, \lambda_d$, training epochs $T$.

**Output:** Trained $\{\mathbf{U}^{(1)}, \mathbf{U}^{(2)}, \boldsymbol{\mathcal{G}}\}$.

**Initialize:** `xavier_uniform`($\{\mathbf{U}^{(1)}, \mathbf{U}^{(2)}, \boldsymbol{\mathcal{G}}\}$).

**for** $t = 1$ ***to*** $T$ **do**

    $\boldsymbol{\mathcal{X}}, \boldsymbol{\mathcal{Y}} \leftarrow$ `sample_batch`($\mathcal{D}$);

    $\hat{\boldsymbol{\mathcal{Y}}} \leftarrow$ `forward`($\boldsymbol{\mathcal{X}}, \{\mathbf{U}^{(1)}, \mathbf{U}^{(2)}, \boldsymbol{\mathcal{G}}\}$) via Eq. 2;

    ▷ *Orthogonal regularization*

    $\mathcal{R}_d(\mathbf{U}^{(1)}) \leftarrow \frac{\rho}{\Phi_1^2}(\|\mathbf{U}^{(1)^{\mathrm{T}}}\mathbf{U}^{(1)} - \mathbf{I}\|_F^2 + \|\mathbf{U}^{(1)}\mathbf{U}^{(1)^{\mathrm{T}}} - \mathbf{I}\|_F^2)$;

    $\mathcal{R}_d(\mathbf{U}^{(2)}) \leftarrow \frac{\rho}{\Phi_2^2}(\|\mathbf{U}^{(2)^{\mathrm{T}}}\mathbf{U}^{(2)} - \mathbf{I}\|_F^2 + \|\mathbf{U}^{(2)}\mathbf{U}^{(2)^{\mathrm{T}}} - \mathbf{I}\|_F^2)$;

    `loss` $\leftarrow \mathcal{L}(\boldsymbol{\mathcal{Y}}, \hat{\boldsymbol{\mathcal{Y}}}) + \lambda_d(\mathcal{R}_d(\mathbf{U}^{(1)}) + \mathcal{R}_d(\mathbf{U}^{(2)}))$;

    `update`($\{\mathbf{U}^{(1)}, \mathbf{U}^{(2)}, \boldsymbol{\mathcal{G}}\}$, `loss`);

**end**

---

To verify this hypothesis, we also perform ablation study for low-tensor-rank training using different orthogonal regularization schemes. As reported in the **Appendix** B, DSO scheme shows consistently better performance than other schemes, and hence it is adopted in our proposed low-rank training procedure.

**Overall Training Procedure.** Based on the above analysis and proposals, we summarize them and develop the corresponding efficient low-rank training (ELRT) algorithm to train high-performance high-compactness low-tensor rank CNN model from scratch. Algorithm 1 describes the details.

## 5 EXPERIMENTS

**Datasets and Baselines.** We evaluate our approach on CIFAR-10 and ImageNet datasets with different inference and training FLOPs reduction ratios (e.g., $2\times$ FLOPs reduction ratios refers to 50% FLOPs reduction). For experiments on CIFAR-10 dataset, the performance of ELRT for VGG-16, ResNet-20, ResNet-56 and MobileNetV2 are evaluated and compared with the results using compression and structured sparse training methods. For experiments on ImageNet dataset, we evaluate our approach for training ResNet-50.

**Calculation of Inference and Training FLOPs Reduction.** A very important benefit provided by compression-aware training, e.g., low-rank training and sparse training, is simultaneously achieving both the "Inference FLOPs Reduction" and "Training FLOPs Reduction". The details of calculation mechanism for these two metrics are described in the **Appendix** C.

**Hyperparameter.** We use SGD optimizer for training with batch size, momentum and weight decay as 128, 0.9 and 0.0001, respectively. The learning rates are set as 0.1 on CIFAR-10 dataset and 0.05

on ImageNet dataset, respectively, with the cosine scheduler. All experiments are performed via using PyTorch 1.12 and following the PyTorch official training strategy. The detailed configurations for the tensor rank values in each layer are reported in the **Appendix** D.

Table 1: Results for VGG-16, ResNet-20, ResNet-56 and MobileNetV2 on CIFAR-10 dataset. "*" denotes compression ratio since the corresponding work does not report FLOPs reduction.

| Method | Always Train Compact Model | Direct Train (No Pre-train) | Post-Train Model | Top-1 (%) | Inference FLOPs ↓ | Training FLOPs ↓ |
|---|---|---|---|---|---|---|
| | | | VGG-16 | | | |
| Original VGG-16 | ✗ | ✓ | Dense | 93.96 | 1.00× | 1× |
| LREL (Idelbayev & Carreira-Perpinán (2020)) | ✗ | ✗ | Low-rank | 92.72 | 6.88× | <1× |
| ALDS (Liebenwein et al. (2021)) | ✓ | ✗ | Low-rank | 92.67 | 7.26× | <1× |
| TETD (Yin et al. (2021)) | ✗ | ✗ | Low-rank | 93.11 | 7.37× | <1× |
| GrowEfficient (Yuan et al. (2021)) | ✓ | ✓ | Sparse | 92.50 | 7.35× | 1.22× |
| BackSparse (Zhou et al. (2021)) | ✓ | ✓ | Sparse | 92.50 | 11.5× | 8.69× |
| HRank (Lin et al. (2020a)) | ✓ | ✗ | Sparse | 92.34 | 2.24× | <1× |
| CHIP (Sui et al. (2021)) | ✓ | ✗ | Sparse | 93.18 | 4.67× | <1× |
| **ELRT** (Ours) | ✓ | ✓ | Low-rank | **93.30** | **9.16×** | **9.16×** |
| **ELRT** (Ours) | ✓ | ✓ | Low-rank | **92.99** | **12.4×** | **12.4×** |
| | | | ResNet-20 | | | |
| Original ResNet-20 | ✗ | ✓ | Dense | 91.25 | 1.00× | 1× |
| SVDT (Yang et al. (2020)) | ✓ | ✗ | Low-rank | 90.39 | 2.94× | <1× |
| PSTRN-S (Li et al. (2021a)) | ✓ | ✗ | Low-rank | 90.80 | 2.25×* | <1× |
| ALDS (Liebenwein et al. (2021)) | ✓ | ✗ | Low-rank | 90.92 | 3.11× | <1× |
| LREL (Idelbayev & Carreira-Perpinán (2020)) | ✗ | ✗ | Low-rank | 90.20 | 3.00× | <1× |
| GrowEfficient (Yuan et al. (2021)) | ✓ | ✓ | Sparse | 90.91 | 2.00× | 1.13× |
| BackSparse (Zhou et al. (2021)) | ✓ | ✓ | Sparse | 90.93 | 2.77× | 2.09× |
| **ELRT** (Ours) | ✓ | ✓ | Low-rank | **91.73** | **1.98×** | **1.98×** |
| **ELRT** (Ours) | ✓ | ✓ | Low-rank | **91.26** | **2.51×** | **2.51×** |
| **ELRT** (Ours) | ✓ | ✓ | Low-rank | **90.95** | **3.02×** | **3.02×** |
| **ELRT** (Ours) | ✓ | ✓ | Low-rank | **89.64** | **6.01×*** | **3.87×** |
| | | | ResNet-56 | | | |
| Original ResNet-56 | ✗ | ✓ | Dense | 93.26 | 1.00× | 1× |
| SVDT (Yang et al. (2020)) | ✓ | ✗ | Low-rank | 93.17 | 3.75× | <1× |
| TRP (Xu et al. (2020)) | ✗ | ✗ | Low-rank | 92.63 | 2.43× | <1× |
| ENC-Inf (Kim et al. (2019)) | ✓ | ✗ | Low-rank | 93.00 | 2.00× | <1× |
| CaP (Minnehan & Savakis (2019)) | ✗ | ✗ | Low-rank | 93.22 | 2.00× | <1× |
| CC (Li et al. (2021b)) | ✓ | ✗ | Low-rank | 93.64 | 2.08× | <1× |
| **ELRT** (Ours) | ✓ | ✓ | Low-rank | **93.96** | **2.05×** | **2.05×** |
| **ELRT** (Ours) | ✓ | ✓ | Low-rank | **94.01** | **2.52×** | **2.52×** |
| **ELRT** (Ours) | ✓ | ✓ | Low-rank | **93.67** | **3.01×** | **3.01×** |
| | | | MobileNetV2 | | | |
| Original MobileNetV2 | ✗ | ✓ | Dense | 94.48 | 1.00× | 1× |
| GAL (Lin et al. (2019)) | ✗ | ✗ | Sparse | 93.07 | 1.69× | <1× |
| DCP (Zhuang et al. (2018)) | ✗ | ✗ | Sparse | 94.25 | 1.36× | <1× |
| SCOP (Tang et al. (2020)) | ✓ | ✗ | Sparse | 94.24 | 1.67× | <1× |
| **ELRT** (Ours) | ✓ | ✓ | Low-rank | **94.87** | **1.47×** | **1.47×** |
| **ELRT** (Ours) | ✓ | ✓ | Low-rank | **94.77** | **1.71×** | **1.71×** |

**VGG-16, ResNet-20, ResNet-56, MobileNetV2 on CIFAR-10.** As shown in Table 1, for ResNet-20 model, the proposed ELRT can bring 0.48% accuracy increase over baseline model with $1.98\times$ inference and training FLOPs reduction. For ResNet-56 model, ELRT method can bring 0.75% accuracy increase over baseline model with $2.52\times$ inference and training FLOPs reduction. We also compare ELRT with several compression approaches on MobileNetV2 model. ELRT brings $1.71\times$ inference and training FLOPs reduction with 0.29% accuracy increase over the baseline. Notice that similar to TETD, HRank and CHIP, we add BN layers in VGG-16 to stable the training process.

**ResNet-50 on ImageNet.** Table 2 summaries the performance of different methods for generating the compact ResNet-50 model on ImageNet. Compared with the state-of-the-art approaches, ELRT achieves at least 0.49% accuracy increase with $2.19\times$ inference and training FLOPs reduction.

**Remark: Benefits of ELRT on Training FLOPs Reduction.** From Table 1 and 2, it is seen that ELRT enjoys two benefits on low training cost. First, unlike low-rank compression or pruning, ELRT can reduce overall training cost because it eliminates the need of pre-training phase. Second, unlike structured sparse training, e.g., GrowEfficient and BackSparse, ELRT has the same FLOPs

reduction in both inference and training. This is because structured sparse training consumes extra computation to calculate the channel mask during backward propagation, which is not needed in low-rank training. More details of calculation of Inference and Training FLOPs reduction for different methods are reported in the **Appendix** C.

Table 2: Results for ResNet-50 on ImageNet dataset.

| Method | Always Train Compact Model | Direct Train (No Pre-train) | Post-Train Model | Top-1 (%) | Top-5 (%) | Inference FLOPs ↓ | Training FLOPs ↓ |
|---|---|---|---|---|---|---|---|
| | | ResNet-50 | | | | | |
| Original ResNet-50 | ✗ | ✓ | Dense | 76.15 | 92.87 | 1.00× | 1× |
| AutoP (Luo & Wu (2020)) | ✗ | ✗ | Sparse | 74.76 | 92.15 | 1.95× | <1× |
| PruneTrain (Lym et al. (2019)) | ✗ | ✓ | Sparse | 74.62 | N/A | 2.13× | 1.67× |
| HRank (Lin et al. (2020a)) | ✓ | ✗ | Sparse | 74.98 | 92.33 | 1.78× | <1× |
| SCOP (Tang et al. (2020)) | ✓ | ✗ | Sparse | 75.26 | 92.53 | 2.21× | <1× |
| CHIP (Sui et al. (2021)) | ✓ | ✗ | Sparse | 75.26 | 92.53 | 2.68× | <1× |
| ISP (Ganjdanesh et al. (2022)) | ✗ | ✗ | Sparse | 75.97 | 92.74 | 2.31× | <1× |
| SVDT (Yang et al. (2020)) | ✓ | ✗ | Low-rank | N/A | 91.81 | 1.79× | <1× |
| TRPNu (Xu et al. (2020)) | ✗ | ✗ | Low-rank | 74.06 | 92.07 | 1.80× | <1× |
| Stable (Phan et al. (2020)) | ✓ | ✗ | Low-rank | 74.68 | 92.16 | 2.64× | <1× |
| Hinge (Li et al. (2020)) | ✗ | ✗ | Low-rank | 74.70 | N/A | 2.15× | <1× |
| CC (Li et al. (2021b)) | ✓ | ✗ | Low-rank | 75.59 | 92.64 | 2.12× | <1× |
| CC (Li et al. (2021b)) | ✓ | ✗ | Low-rank | 74.54 | 92.25 | 2.68× | <1× |
| GrowEfficient (Yuan et al. (2021)) | ✓ | ✓ | Sparse | 75.20 | N/A | 1.99× | 1.10× |
| BackSparse (Zhou et al. (2021)) | ✓ | ✓ | Sparse | 76.00 | N/A | 2.13× | 1.60× |
| **ELRT** (Ours) | ✓ | ✓ | Low-rank | **76.49** | **93.28** | **2.19×** | **2.19×** |
| **ELRT** (Ours) | ✓ | ✓ | Low-rank | **76.11** | **92.88** | **2.49×** | **2.49×** |
| **ELRT** (Ours) | ✓ | ✓ | Low-rank | **75.32** | **92.57** | **2.91×** | **2.91×** |

**Comparison with Existing Low-rank Training and Directly Training Small Model.** Because the existing low-rank training works (Tai et al. (2015); Ioannou et al. (2015); Hawkins et al. (2022)) are evaluated on small dataset (MNIST) and/or non-popularly used models, we report their comparison with ELRT in the **Appendix** A.3. Also, we compare ELRT with directly training a small dense models with the same target model size, and the results are reported in the **Appendix** A.4.

**Ablation Study.** We perform ablation study on the effect of imposing orthogonality and different orthogonal regulation schemes, the details are reported in the **Appendix** B.

**Practical Speedup of Low-Tensor-Rank CNNs.** To demonstrate the practical effectiveness of low-tensor-rank models, we measure the inference time of the models trained by using ELRT. Here we evaluate the speedup on four hardware platforms: Nvidia V100 (desktop GPU), Nvidia Jetson TX2 (embedded GPU), Xilinx PYNQZ1 (FPGA) and Eyeriss (ASIC). The evaluated models are low-rank ResNet-50 for ImageNet with three target FLOPs reduction settings. As summarized in Table 3, the low-rank CNNs obtained from ELRT enjoy considerable speedup across different hardware platforms.

Table 3: Runtime (per image) for the low-rank ResNet-50 trained via our proposed ELRT.

| | FLOPs Reduction | ASIC Eyeriss 65nm | FPGA Xilinx PYNQZ1 | Desktop GPU Nvidia V100 | Embedded GPU Nvidia Jetson TX2 |
|---|---|---|---|---|---|
| Original ResNet-50 | 1× | 38.10ms (1×) | 172.0ms (1×) | 0.8145ms (1×) | 29.46ms (1×) |
| **ELRT** (Ours) | 2.19× | **19.24ms (1.98×)** | **85.62ms (2.01×)** | **0.6934ms (1.17×)** | **21.47ms (1.37×)** |
| **ELRT** (Ours) | 2.49× | **15.96ms (2.39×)** | **71.08ms (2.42×)** | **0.6241ms (1.31×)** | **19.24ms (1.53×)** |
| **ELRT** (Ours) | 2.91× | **13.85ms (2.75×)** | **61.02ms (2.82×)** | **0.5963ms (1.37×)** | **18.29ms (1.61×)** |

## 6 CONCLUSION

This paper proposes ELRT, an efficient low-rank training approach for training CNN models from scratch. ELRT is essentially an orthogonality-aware training solution that can impose and ensure the important orthogonality on the low-rank models during the training process. Evaluation results demonstrate the effectiveness of ELRT as compared to state-of-the-art model compression and other compression-aware training approaches.

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

APPENDIX

The entire Appendix consists of four sections.

- Section A lists the additional experiments results with more model types and more comparisons.

- Section B reports the ablation study for the orthogonality-imposing strategy.

- Section C presents the calculation schemes for two important performance metrics: "inference FLOPs reduction" and "training FLOPs reduction".

- Section D shows the layer-wise rank distribution in the experiments, thereby demonstrating the convenience of rank setting.

## A  ADDITIONAL EXPERIMENTS

### A.1  RESNET-32 AND WRN-28-8 ON CIFAR-10

Table 4 shows the evaluation results of low-rank training for ResNet-32 and WideResNet-28-8 model on CIFAR-10 dataset. Ror ResNet-32 mode, it is seen that the proposed ELRT method can bring 0.23% accuracy increase over the baseline model with 2.31× inference and training FLOPs reduction. Compared with the existing methods, the proposed low-rank training from scratch approach shows better performance with respect to FLOPs reduction and accuracy.

Table 4: Results for ResNet-32 and WRN-28-8 on CIFAR-10 dataset. "∗" denotes compression ratio since the corresponding work does not report FLOPs reduction.

| Method | Always Train Compact Model | Direct Train (No Pre-train) | Post-Train Model | Top-1 (%) | Inference FLOPs ↓ | Training FLOPs ↓ |
|---|---|---|---|---|---|---|
| | | ResNet-32 | | | | |
| Original ResNet-32 | ✗ | ✓ | Dense | 92.49 | 1.00× | 1× |
| FPGM(He et al. (2019)) | ✓ | ✗ | Sparse | 91.93 | 2.12× | <1× |
| SCOP(Tang et al. (2020)) | ✓ | ✗ | Sparse | 92.13 | 2.27× | <1× |
| SVDT(Yang et al. (2020)) | ✓ | ✗ | Low-rank | 90.55 | 3.93× | N/A |
| PSTRN-S(Li et al. (2021a)) | ✓ | ✗ | Low-rank | 91.44 | 2.60×* | <1× |
| **ELRT** (Ours) | ✓ | ✓ | Low-rank | **92.67** | **2.05×** | **2.05×** |
| **ELRT** (Ours) | ✓ | ✓ | Low-rank | **92.72** | **2.31×** | **2.31×** |
| **ELRT** (Ours) | ✓ | ✓ | Low-rank | **92.03** | **3.01×** | **3.01×** |
| **ELRT** (Ours) | ✓ | ✓ | Low-rank | **91.21** | **5.40×*** | **3.34×** |
| | | WRN-28-8 | | | | |
| Original WRN-28-8 | ✗ | ✓ | Dense | 95.60 | 1.00× | 1× |
| DST (Liu et al. (2020)) | ✓ | ✓ | Sparse | 94.80 | 10.0×* | N/A |
| SFP (He et al. (2018)) | ✓ | ✓ | Sparse | 94.22 | 5.00×* | <1× |
| DPF (Lin et al. (2020c)) | ✓ | ✓ | Sparse | 95.15 | 5.00×* | N/A |
| **ELRT** (Ours) | ✓ | ✓ | Low-rank | **95.65** | **5.36×*** | **3.14×** |
| **ELRT** (Ours) | ✓ | ✓ | Low-rank | **95.22** | **6.44×*** | **3.32×** |
| **ELRT** (Ours) | ✓ | ✓ | Low-rank | **94.87** | **11.8×*** | **5.76×** |

### A.2  RESNET-18 ON IMAGENET

Table 5 shows the evaluation results of low-rank training for ResNet-18 on ImageNet dataset. It is seen that the proposed ELRT method can increase 0.01% accuracy over the baseline model with 1.40× inference and training FLOPs reduction.

Table 5: Results for ResNet-18 on ImageNet dataset.

| Method | Always Train Compact Model | Direct Train (No Pre-train) | Post-Train Model | Top-1 (%) | Top-5 (%) | Inference FLOPs ↓ | Training FLOPs ↓ |
|---|---|---|---|---|---|---|---|
| | | ResNet-18 | | | | | |
| Original ResNet-18 | ✗ | ✓ | Dense | 69.79 | 89.08 | 1.00× | 1× |
| FPGM (He et al. (2019)) | ✓ | ✗ | Sparse | 68.34 | 88.53 | 1.72× | <1× |
| DSA (Ning et al. (2020)) | ✓ | ✗ | Sparse | 68.61 | 88.35 | 1.67× | <1× |
| SCOP (Tang et al. (2020)) | ✓ | ✗ | Sparse | 68.62 | 88.45 | 1.81× | <1× |
| SVDT (Yang et al. (2020)) | ✓ | ✗ | Low-rank | N/A | 87.26 | 2.03× | <1× |
| TRP (Xu et al. (2020)) | ✗ | ✗ | Low-rank | 65.46 | 86.48 | 1.81× | <1× |
| **ELRT** (Ours) | ✓ | ✓ | Low-rank | **69.80** | **89.29** | **1.40×** | **1.40×** |
| **ELRT** (Ours) | ✓ | ✓ | Low-rank | **68.75** | **88.61** | **1.84×** | **1.84×** |
| **ELRT** (Ours) | ✓ | ✓ | Low-rank | **68.22** | **88.18** | **2.17×** | **2.17×** |

## A.3 COMPARISON WITH EXISTING LOW-RANK TRAINING

We also compare the performance of ELRT with the existing low-rank training methods (Tai et al. (2015); Ioannou et al. (2015); Hawkins et al. (2022)). Table 6 shows the experimental results of training NIN model (Lin et al. (2013)) from scratch on CIFAR-10 dataset. It is seen that compared with (Tai et al. (2015); Ioannou et al. (2015)), ELRT can achieve higher FLOPs reduction with providing better accuracy performance. In addition, we also compare our proposed method with another Tucker-format work (Hawkins et al. (2022)) for low-rank training from scratch. As shown in Table 7, ELRT can achieve higher accuracy and FLOPs reduction than (Hawkins et al. (2022)). In addition, consider ELRT does not require computation-intensive Bayesian estimation that is used in (Hawkins et al. (2022)), ELRT is more attractive for practical applications.

Table 6: Comparison with existing low-rank training works for NIN on CIFAR-10 dataset.

| Method | Always Train Compact Model | Direct Train (No Pre-train) | Post-Train Model | Top-1 (%) | Inference FLOPs ↓ | Training FLOPs ↓ |
|---|---|---|---|---|---|---|
| | | NIN | | | | |
| Original NIN | ✗ | ✓ | Dense | 91.88 | 1.00× | 1× |
| LRR (Tai et al. (2015)) | ✓ | ✓ | Low-rank | 93.02 | 1.50× | N/A |
| LRF (Ioannou et al. (2015)) | ✓ | ✓ | Low-rank | 91.78 | 1.86× | N/A |
| **ELRT** (Ours) | ✓ | ✓ | Low-rank | **93.16** | **1.62×** | **1.62×** |
| **ELRT** (Ours) | ✓ | ✓ | Low-rank | **92.23** | **2.03×** | **2.03×** |
| **ELRT** (Ours) | ✓ | ✓ | Low-rank | **91.86** | **2.52×** | **2.52×** |

Table 7: Results for low-rank training works on MNIST dataset. The model used here can be referred to (Hawkins et al. (2022)).

| Method | Always Train Compact Model | Direct Train (No Pre-train) | Post-Train Model | Top-1 (%) | Inference FLOPs ↓ | Training FLOPs ↓ |
|---|---|---|---|---|---|---|
| | | Customized Model from (Hawkins et al. (2022)) | | | | |
| Original Customized Model | ✗ | ✓ | Dense | 98.09 | 1.00× | 1× |
| ARD-LU (Hawkins et al. (2022)) | ✓ | ✓ | Low-rank | 98.30 | 4.00× | N/A |
| ARD-HC (Hawkins et al. (2022)) | ✓ | ✓ | Low-rank | 98.30 | 4.50× | N/A |
| **ELRT** (Ours) | ✓ | ✓ | Low-rank | **98.41** | **4.74×** | **4.74×** |

## A.4 COMPARISON WITH OTHER COMPACT MODELS TRAINED FROM SCRATCH

We also compare the performance of ELRT with directly training small dense models from scratch with the similar target model sizes. Table 8 shows the experimental results of training ResNet-20 model from scratch on CIFAR-10 dataset. It is seen that compared with models by uniformly removing 25% and 37.5% filters, ELRT can achieve better FLOPs reduction with providing higher accuracy.

Table 8: Comparison with directly training small dense models with the similar target model sizes on CIFAR-10 dataset.

| Method | Always Train Compact Model | Direct Train (No Pre-train) | Post-Train Model | Top-1 (%) | Inference FLOPs ↓ | Training FLOPs ↓ |
|---|---|---|---|---|---|---|
| | | ResNet-20 | | | | |
| Original ResNet-20 | ✗ | ✓ | Dense | 91.25 | 1.00× | 1× |
| Uniform-0.75 | ✓ | ✓ | Dense | 91.08 | 1.70× | 1.70× |
| **ELRT** (Ours) | ✓ | ✓ | Low-rank | **91.73** | **1.98×** | **1.98×** |
| Uniform-0.625 | ✓ | ✓ | Dense | 90.11 | 2.49× | 2.49× |
| **ELRT** (Ours) | ✓ | ✓ | Low-rank | **91.26** | **2.51×** | **2.51×** |

## A.5 COMPARISON WITH OTHER RELATED WORKS

We also compare the performance of ELRT with (1) the works that employ tensor networks straightforwardly (Hayashi et al. (2019); Su et al. (2022)); (2) the works that are trained densely, and compressed with low-rank tensors (Lin et al. (2020b); Yu et al. (2021)); (3) other advanced low-rank schemes (Khodak et al. (2020); Wang et al. (2021); Waleffe & Rekatsinas (2020)).

(Hayashi et al. (2019)) explores to train low-rank CNNs with new discovered decomposition formats via using architecture search. A key drawback is its very high training cost. As reported in (Hayashi et al. (2019)), it needs 829 GPU days to train ResNet-50 on CIFAR-10 dataset, limiting the practicality of this approach, especially on large-scale dataset. Instead, ELRT provides an efficient low-rank training solution with much fewer costs and higher model performance. As shown in the table below, ELRT shows higher 0.5% higher accuracy than (Hayashi et al. (2019)) with 2 times fewer model parameters for training low-rank ResNet-50 on CIFAR-10 dataset.

Table 9: Comparison with Hayashi et al. (2019) on CIFAR-10 dataset.

| Method | Remaining Parameters | Top-1 (%) |
|---|---|---|
| | ResNet-50 | |
| (Hayashi et al. (2019)) | 22.2M | 92.2 |
| ELRT (Ours) | 10.6M | 92.73 |

Similar to (Hayashi et al. (2019)), (Su et al. (2022)) also aims to train low-rank CNNs using new decomposition formats. However, the model obtained via using the method proposed in [R2] has limited performance. As shown in the following Table, for training low-rank ResNet-32 on CIFAR-10 dataset, ELRT shows much higher accuracy (at least 3%) than (Su et al. (2022)) with even higher compression ratio.

Table 10: Comparison with Su et al. (2022) on CIFAR-10 dataset.

| Method | Compression Ratio | Top-1 (%) |
|---|---|---|
| | ResNet-32 | |
| mCP (Su et al. (2022)) | 10.0× | 82.93 |
| mTK (Su et al. (2022)) | 10.0× | 65.75 |
| mTT (Su et al. (2022)) | 10.0× | 83.08 |
| ELRT (Ours) | 10.2× | 87.89 |
| ELRT (Ours) | 12.7× | 86.60 |

(Lin et al. (2020b)) is built on generalized higher-order Tucker articulated kernel scheme. A key difference between (Lin et al. (2020b)) and ELRT is that (Lin et al. (2020b)) aims for low-rank com-

pression that requires the existence of a pre-trained model; while ELRT perform low-rank training from scratch without consuming any pre-training cost, significantly reducing training complexity. More importantly, as shown in the following table, ELRT shows better model performance (more than 1% accuracy increase) than (Lin et al. (2020b)) for obtaining low-rank AlexNet on CIFAR-10 dataset. Here the ranks for Conv2, Conv3, Conv4 and Conv5 are set as [32, 64], [72, 108], [64, 32] and [40, 40], respectively.

Table 11: Comparison with Lin et al. (2020b) on CIFAR-10 dataset.

| Method | Compression Ratio | Top-1 (%) |
|---|---|---|
| AlexNet | | |
| Hotcake (Lin et al. (2020b)) | 9.4× | 83.17 |
| ELRT (Ours) | 9.4× | 84.45 |

(Yu et al. (2021)) is also a low-rank compression work that requiring pre-training phase. As shown in the following table, even without using any pre-trained high-accuracy model, ELRT still achieves higher accuracy than the pre-trained model-required (Yu et al. (2021)) for obtaining low-rank ResNet-56 on CIFAR-10 and ResNet-50 on ImageNet.

Table 12: Comparison with Yu et al. (2021).

| Method | FLOPs Reduction ↓ | Top-1 (%) |
|---|---|---|
| ResNet-56 | | |
| (Yu et al. (2021)) | 2.8× | 93.52 |
| ELRT (Ours) | 3.0× | 93.67 |
| ResNet-50 | | |
| (Yu et al. (2021)) | 2.5× | 75.58 |
| ELRT (Ours) | 2.5× | 76.11 |

(Khodak et al. (2020)) applies spectral initialization (SI) and Frobenius decay (FD) for low-rank training. As shown in the following table, ELRT outperforms different variants of [R5] when training VGG on CIFAR-10 dataset with higher compression ratio and at least 1.4% accuracy increase.

Table 13: Comparison with Khodak et al. (2020) on CIFAR-10 dataset.

| Method | Compression Ratio | Top-1 (%) |
|---|---|---|
| VGGNet | | |
| Low-rank (Khodak et al. (2020)) | 10.0× | 90.71 |
| Low-rank+FI (Khodak et al. (2020)) | 10.0× | 90.99 |
| Low-rank+FD (Khodak et al. (2020)) | 10.0× | 91.57 |
| Low-rank+SI+FD (Khodak et al. (2020)) | 10.0× | 91.58 |
| ELRT (Ours) | 12.4× | 92.99 |

(Wang et al. (2021)) needs a warm-up phase to train a full-rank model in the first epochs, so it does not reduce the overall memory requirement, which is measured by the peak memory usage. Instead, ELRT trains a low-rank model from scratch and always keeps the low-rank format during the training. As shown in the following table, ELRT brings higher model accuracy with fewer FLOPs than (Wang et al. (2021)) for training ResNet-50 on ImageNet.

Table 14: Comparison with Wang et al. (2021) on ImageNet dataset.

| Method | FLOPs Reduction ↓ | Top-1 (%) |
|---|---|---|
| ResNet-50 | | |
| FP32 (Wang et al. (2021)) | 1.14× | 76.43 |
| AMP (Wang et al. (2021)) | 1.14× | 76.35 |
| ELRT (Ours) | 2.2× | 76.49 |

Similar to (Wang et al. (2021)), (Waleffe & Rekatsinas (2020)) is initialized with a wide and full-rank model and then decompose it low-rank format after a few epochs. Instead, ELRT performs low-rank training from scratch and always keeps the model stay in the low-rank format during the entire training procedure. As shown in the following table, ELRT shows 0.51% accuracy increase over (Waleffe & Rekatsinas (2020)) with 2 times model size reduction for training ResNet-50 on ImageNet dataset.

Table 15: Comparison with Waleffe & Rekatsinas (2020) on ImageNet dataset.

| Method | Remaining Parameters | Top-1 (%) |
|---|---|---|
| ResNet-50 | | |
| (Waleffe & Rekatsinas (2020)) | 25.5M | 75.6 |
| ELRT (Ours) | 11.4M | 76.11 |

## B    ABLATION STUDIES

**Effect of Imposing Orthogonality.** We conduct experiments to study the effect of imposing the orthogonality on the factor matrices via using DSO regularization. As shown in Fig. 6, our proposed orthogonality-aware low-rank training shows very significant performance improvement than the standard low-rank training with the same FLOPs reduction and same low-rank format, thereby demonstrating the importance of enforcing orthogonality on the components of the low-rank model.

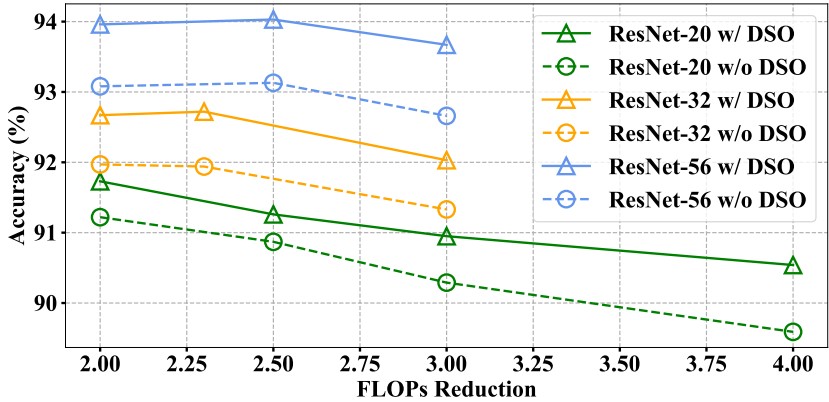

Figure 6: Performance of low-tensor-rank training with and without using DSO for ResNet-20/32/56 on CIFAR-10 dataset.

**Different Orthogonal Regularization Schemes.** We also conduct the ablation study to explore the best-suited scheme to impose the orthogonality. As shown in Table 16, DSO regularization demon-

strates consistently better performance than the other schemes. Such empirical results also coincide with our analysis in **Question 3**. Therefore we adopt the DSO scheme for all our experiments.

Table 16: Performance of using different orthogonal regularization schemes for training low-rank ResNet-20, ResNet-32 and ResNet-56 on CIFAR-10 dataset.

| FLOPs ↓ | SO | SRIP | MC | DSO |
|---|---|---|---|---|
| | | ResNet-20 | | |
| 2× | 91.43 | 91.32 | 91.36 | **91.73** |
| 2.5× | 91.05 | 90.92 | 90.91 | **91.26** |
| 3× | 90.72 | 90.59 | 90.43 | **90.95** |
| | | ResNet-32 | | |
| 2× | **92.75** | 92.38 | 92.41 | 92.67 |
| 3× | 91.79 | 91.72 | 91.76 | **92.03** |
| | | ResNet-56 | | |
| 2× | 93.52 | 93.44 | 93.45 | **93.96** |
| 3× | 93.35 | 93.26 | 93.28 | **93.67** |

**Effect of Different Orthogonality Penalty Parameters** $\lambda_d$**.** As shown in Table 17, 18 and Figure 7, we evaluate the effect of different orthogonality penalty parameters $\lambda_d$.

Table 17: ResNet-20 with different orthogonality penalty parameters ($\lambda_d$).

| $\lambda_d$ | FLOPs ↓ | Accuracy (%) | FLOPs ↓ | Accuracy (%) | FLOPs ↓ | Accuracy (%) |
|---|---|---|---|---|---|---|
| 1e-6 | | 91.48 | | 91.11 | | 90.68 |
| 5e-4 | | 91.64 | | 91.16 | | 90.80 |
| 1e-3 | | 91.73 | | 91.26 | | 90.95 |
| 5e-3 | 1.98× | 91.61 | 2.51× | 91.29 | 3.02× | 90.96 |
| 1e-2 | | 91.66 | | 91.22 | | 90.88 |
| 5e-2 | | 91.55 | | 91.37 | | 91.28 |
| 1 | | 91.78 | | 91.12 | | 91.09 |

Table 18: ResNet-56 with different orthogonality penalty parameters ($\lambda_d$).

| $\lambda_d$ | FLOPs ↓ | Accuracy (%) | FLOPs ↓ | Accuracy (%) | FLOPs ↓ | Accuracy (%) |
|---|---|---|---|---|---|---|
| 1e-6 | | 92.82 | | 93.24 | | 93.28 |
| 5e-4 | | 93.62 | | 93.86 | | 93.51 |
| 1e-3 | | 93.96 | | 94.01 | | 93.67 |
| 5e-3 | 2.05× | 93.95 | 2.52× | 93.87 | 3.01× | 93.64 |
| 1e-2 | | 93.87 | | 93.86 | | 93.59 |
| 5e-2 | | 94.03 | | 93.71 | | 93.55 |
| 1 | | 93.87 | | 93.75 | | 93.64 |

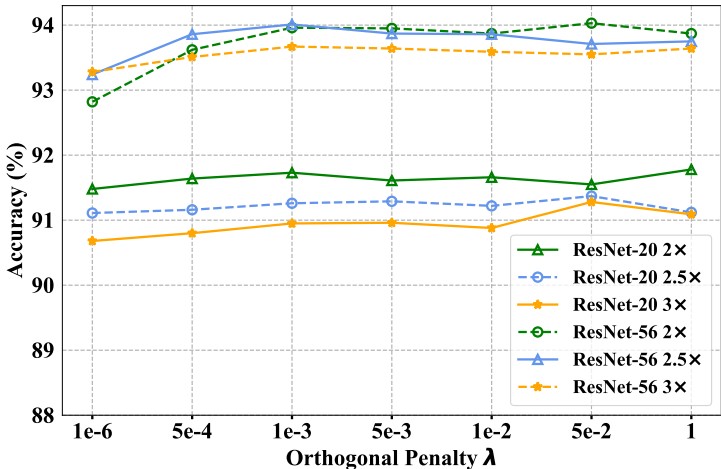

Figure 7: Performance of ELRT for ResNet-20/56 on CIFAR-10 dataset with different orthogonal penalty parameters $\lambda_d$.

## C CALCULATION SCHEMES FOR INFERENCE AND TRAINING FLOPS REDUCTION

### C.1 CALCULATION OF INFERENCE FLOPS REDUCTION

To calculate the inference FLOPs reduction of one convolution layer after using Tucker-2 decomposition, we adopt the scheme used in (Kim et al. (2015)):

$$E = \frac{D^2 S T H' W'}{S R_1 H' W' + D^2 R_1 R_2 H' W' + T R_2 H' W'}, \tag{7}$$

where $D$ is the kernel height and width, $S$ is the number of input channels, $T$ is the number of output channels, $H', W'$ are output height and width, and $R_1, R_2$ are the ranks of the Tucker decomposition.

With the above notation, the inference FLOPs reduction can be calculated as $\frac{\sum_{\text{all layers}} A}{\sum_{\text{all layers}} B}$, where $A = D^2 S T H' W'$ represents the number of multiplication-addition operations of a layer in the oiginal model, and $B = S R_1 H' W' + D^2 R_1 R_2 H' W' + T R_2 H' W'$ represents the number of multiplication-addition operations of a layer in the Tucker decomposed low-rank model.

### C.2 CALCULATION OF TRAINING FLOPS REDUCTION

As defined in (Zhou et al. (2021); Evci et al. (2020)), "training FLOPs reduction", also noted as "training-cost saving", is the ratio of the average FLOPs of the dense network over that of the compact network. Here the total FLOPs of training a network consists of the part in forward pass and backward pass. As indicated in (Evci et al. (2020); Zhou et al. (2021); Baydin et al. (2018)), the FLOPs of backward propagation can be roughly counted as about 2 times of that is consumed in the forward propagation. Next we denote the FLOPs of the dense network, the sparse network, the low-rank network during forward propagation as $f_D, f_S$ and $f_L$, respectively.

### C.2.1 DENSE NETWORK.

The FLOPs of the forward propagation is $f_D$. The FLOPs of the backward propagation is $2f_D$. The training FLOPs reduction is $\frac{f_D + 2f_D}{f_D + 2f_D} = 1$, which means there is no training FLOPs reduction.

### C.2.2 PRUNING.

Assume $T$ epochs are needed to obtain a pre-trained model. After pruning, another $K$ epochs are needed to re-train the model for fine-tuning. In the pre-training phase, the total computation cost of the forward propagation and backward propagation are $f_D * T$ and $2f_D * T$, respectively. In the re-training phase, the total computation cost of the forward propagation and backward propagation are $f_S * K$ and $2f_S * K$, respectively. So overall the training FLOPs reduction is $\frac{3f_D * T}{3f_D * T + 3f_S * K} < 1$, which means there is no training FLOPs reduction.

### C.2.3 LOW-RANK COMPRESSION.

Assume $T$ epochs are needed to obtain a pre-trained model. After low-rank decomposition, another $K$ epochs are needed to re-train the model for fine-tuning. In the pre-training phase, the total computation cost of the forward propagation and backward propagation are $f_D * T$ and $2f_D * T$, respectively. In the re-training phase, the total computation cost of the forward propagation and backward propagation are $f_L * K$ and $2f_L * K$, respectively. So overall the training FLOPs reduction is $\frac{3f_D * T}{3f_D * T + 3f_L * K} < 1$, which means there is no training FLOPs reduction.

### C.2.4 STRUCTURED SPARSE TRAINING.

- **GrowEfficient (Yuan et al. (2021)).** For training a sparse model via GrowEfficient, the FLOPs of the forward propagation is $f_S$. Since the channel masks and scores, which determine to keep or remove the corresponding channels/filters, are updated by the Straight-Through gradient Estimation (STE), the backward propagation has to go through all the channels/filters, leading to dense computation. Therefore, the FLOPs of the backward propagation is $2f_D$. For training a dense model, the computation cost of the forward propagation and backward propagation are $f_D$ and $2f_D$, respectively. Therefore, the training FLOPs reduction is $\frac{f_D + 2f_D}{f_S + 2f_D} = \frac{3f_D}{f_S + 2f_D}$.

- **SparseBackward (Zhou et al. (2021)).** For training a sparse model via SparseBackward, the FLOPs of the forward propagation is $f_S$. The backward propagation keeps sparse with $2f_S$ computation cost. Unlike updating masks/scores via dense backward propagation in GrowEfficient, SparseBackward updates them via Variance Reduced Policy Gradient Estimator (VR-PGE), which only requires an extra one-time forward propagation with training cost $f_S$. For training a dense model, the computation cost of the forward propagation and backward propagation are $f_D$ and $2f_D$, respectively. Therefore, the training FLOPs reduction is $\frac{f_D + 2f_D}{2f_S + 2f_S} = \frac{3f_D}{4f_S}$.

### C.2.5 ELRT (OURS).

Recall that ELRT is built on Tucker-2 decomposition that converts one convolutional layer into two factor matrices, which can be viewed as $1 \times 1$ convolutional layers, and one core tensor that is viewed as $3 \times 3$ convolutional layer. Therefore, our low-rank model can be viewed as a compact dense network with more convolutional layers but fewer FLOPs and parameters. Therefore, assume the FLOPs of the forward propagation is $f_L$. The FLOPs of the backward propagation is $2f_L$. The training FLOPs reduction is calculated as $\frac{f_D + 2f_D}{f_L + 2f_L} = \frac{f_D}{f_L}$, which is identical to the inference FLOPs reduction as calculated in Eq. 7. **In other words, the inference and training FLOPs reductions brought by ELRT are the same.** Notice that the FLOPs of calculating the orthogonality loss term is ignored here since it only occurs per batch, while $f_L$ and $f_D$ are consumed per data. Considering the batch size is typically large (e.g., 128 in our experiment), the FLOPs contribution of calculating the orthogonality loss term is negligible.

## D RANK SETTINGS

Table 19, 20 and 21 list the layer-wise rank settings of ResNet-20 model on CIFAR-10 dataset under $1.98\times$, $3.02\times$ FLOPs and $6.01\times$ parameters reduction, respectively. Table 22, 23 list the layer-wise rank settings of ResNet-56 model on CIFAR-10 dataset under $2.05\times$ and $2.52\times$ FLOPs reduction.

Table 24 lists the layer-wise rank settings of ResNet-50 model on ImageNet dataset under $2.49\times$ FLOPs reduction.

From these tables, it is seen that many adjacent layers share the same rank value, and the entire model only needs a few rank values to be assigned. For instance, for ResNet-56 model under $2.05\times$ FLOPs reduction on CIFAR-10 dataset, only three numbers: 12, 18, 26, are selected as the ranks to be assigned to all layers, where the first/second/last eighteen layers share the rank value 12/18/26, respectively. Such a rank-sharing phenomenon significantly simplifies the rank selection process.

Table 19: Layer-wise rank settings of the compressed ResNet-20 model on CIFAR-10 dataset with $1.98\times$ FLOPs reduction.

| Layer Name | Rank | Layer Name | Rank | Layer Name | Rank |
|---|---|---|---|---|---|
| FLOPs Reduction = 1.98× | | | | | |
| layer1.0.conv1 | (14, 14) | layer2.0.conv1 | (14, 14) | layer3.0.conv1 | (28, 28) |
| layer1.0.conv2 | (12, 12) | layer2.0.conv2 | (14, 14) | layer3.0.conv1 | (28, 28) |
| layer1.1.conv1 | (12, 12) | layer2.1.conv1 | (14, 14) | layer3.1.conv1 | (28, 28) |
| layer1.1.conv2 | (12, 12) | layer2.1.conv2 | (14, 14) | layer3.1.conv1 | (28, 28) |
| layer1.2.conv1 | (12, 12) | layer2.2.conv1 | (16, 16) | layer3.2.conv1 | (28, 28) |
| layer1.2.conv2 | (12, 12) | layer2.2.conv2 | (16, 16) | layer3.2.conv1 | (28, 28) |

Table 20: Layer-wise rank settings of the compressed ResNet-20 model on CIFAR-10 dataset with $3.02\times$ FLOPs reduction.

| Layer Name | Rank | Layer Name | Rank | Layer Name | Rank |
|---|---|---|---|---|---|
| FLOPs Reduction = 3.02× | | | | | |
| layer1.0.conv1 | (10, 10) | layer2.0.conv1 | (12, 12) | layer3.0.conv1 | (20, 20) |
| layer1.0.conv2 | (10, 10) | layer2.0.conv2 | (12, 12) | layer3.0.conv1 | (20, 20) |
| layer1.1.conv1 | (10, 10) | layer2.1.conv1 | (12, 12) | layer3.1.conv1 | (20, 20) |
| layer1.1.conv2 | (8, 8) | layer2.1.conv2 | (14, 14) | layer3.1.conv1 | (22, 22) |
| layer1.2.conv1 | (8, 8) | layer2.2.conv1 | (14, 14) | layer3.2.conv1 | (22, 22) |
| layer1.2.conv2 | (8, 8) | layer2.2.conv2 | (14, 14) | layer3.2.conv1 | (22, 22) |

Table 21: Layer-wise rank settings of the compressed ResNet-20 model on CIFAR-10 dataset with $6.01\times$ parameters reduction.

| Layer Name | Rank | Layer Name | Rank | Layer Name | Rank |
|---|---|---|---|---|---|
| FLOPs Reduction = 6.01× | | | | | |
| layer1.0.conv1 | (9, 9) | layer2.0.conv1 | (12, 12) | layer3.0.conv1 | (16, 16) |
| layer1.0.conv2 | (9, 9) | layer2.0.conv2 | (12, 12) | layer3.0.conv1 | (16, 16) |
| layer1.1.conv1 | (9, 9) | layer2.1.conv1 | (12, 12) | layer3.1.conv1 | (16, 16) |
| layer1.1.conv2 | (9, 9) | layer2.1.conv2 | (12, 12) | layer3.1.conv1 | (16, 16) |
| layer1.2.conv1 | (9, 9) | layer2.2.conv1 | (12, 12) | layer3.2.conv1 | (16, 16) |
| layer1.2.conv2 | (9, 9) | layer2.2.conv2 | (12, 12) | layer3.2.conv1 | (16, 16) |

Table 22: Layer-wise rank settings of the compressed ResNet-56 model on CIFAR-10 dataset with 2.05× FLOPs reduction.

| Layer Name | Rank | Layer Name | Rank | Layer Name | Rank |
|---|---|---|---|---|---|
| FLOPs Reduction = 2.05× | | | | | |
| layer1.0.conv1 | (12, 12) | layer2.0.conv1 | (18, 18) | layer3.0.conv1 | (26, 26) |
| layer1.0.conv2 | (12, 12) | layer2.0.conv2 | (18, 18) | layer3.0.conv1 | (26, 26) |
| layer1.1.conv1 | (12, 12) | layer2.1.conv1 | (18, 18) | layer3.1.conv1 | (26, 26) |
| layer1.1.conv2 | (12, 12) | layer2.1.conv2 | (18, 18) | layer3.1.conv1 | (26, 26) |
| layer1.2.conv1 | (12, 12) | layer2.2.conv1 | (18, 18) | layer3.2.conv1 | (26, 26) |
| layer1.2.conv2 | (12, 12) | layer2.2.conv2 | (18, 18) | layer3.2.conv1 | (26, 26) |
| layer1.3.conv1 | (12, 12) | layer2.3.conv1 | (18, 18) | layer3.3.conv1 | (26, 26) |
| layer1.3.conv2 | (12, 12) | layer2.3.conv2 | (18, 18) | layer3.3.conv1 | (26, 26) |
| layer1.4.conv1 | (12, 12) | layer2.4.conv1 | (18, 18) | layer3.4.conv1 | (26, 26) |
| layer1.4.conv2 | (12, 12) | layer2.4.conv2 | (18, 18) | layer3.4.conv1 | (26, 26) |
| layer1.5.conv1 | (12, 12) | layer2.5.conv1 | (18, 18) | layer3.5.conv1 | (26, 26) |
| layer1.5.conv2 | (12, 12) | layer2.5.conv2 | (18, 18) | layer3.5.conv1 | (26, 26) |
| layer1.6.conv1 | (12, 12) | layer2.6.conv1 | (18, 18) | layer3.6.conv1 | (26, 26) |
| layer1.6.conv2 | (12, 12) | layer2.6.conv2 | (18, 18) | layer3.6.conv1 | (26, 26) |
| layer1.7.conv1 | (12, 12) | layer2.7.conv1 | (18, 18) | layer3.7.conv1 | (26, 26) |
| layer1.7.conv2 | (12, 12) | layer2.7.conv2 | (18, 18) | layer3.7.conv1 | (26, 26) |
| layer1.8.conv1 | (12, 12) | layer2.8.conv1 | (18, 18) | layer3.8.conv1 | (26, 26) |
| layer1.8.conv2 | (12, 12) | layer2.8.conv2 | (18, 18) | layer3.8.conv1 | (26, 26) |

Table 23: Layer-wise rank settings of the compressed ResNet-56 model on CIFAR-10 dataset with 2.52× FLOPs reduction.

| Layer Name | Rank | Layer Name | Rank | Layer Name | Rank |
|---|---|---|---|---|---|
| FLOPs Reduction = 2.52× | | | | | |
| layer1.0.conv1 | (10, 10) | layer2.0.conv1 | (15, 15) | layer3.0.conv1 | (28, 28) |
| layer1.0.conv2 | (10, 10) | layer2.0.conv2 | (15, 15) | layer3.0.conv1 | (28, 28) |
| layer1.1.conv1 | (10, 10) | layer2.1.conv1 | (15, 15) | layer3.1.conv1 | (28, 28) |
| layer1.1.conv2 | (10, 10) | layer2.1.conv2 | (15, 15) | layer3.1.conv1 | (28, 28) |
| layer1.2.conv1 | (10, 10) | layer2.2.conv1 | (15, 15) | layer3.2.conv1 | (28, 28) |
| layer1.2.conv2 | (10, 10) | layer2.2.conv2 | (15, 15) | layer3.2.conv1 | (28, 28) |
| layer1.3.conv1 | (10, 10) | layer2.3.conv1 | (15, 15) | layer3.3.conv1 | (28, 28) |
| layer1.3.conv2 | (10, 10) | layer2.3.conv2 | (15, 15) | layer3.3.conv1 | (28, 28) |
| layer1.4.conv1 | (10, 10) | layer2.4.conv1 | (15, 15) | layer3.4.conv1 | (28, 28) |
| layer1.4.conv2 | (10, 10) | layer2.4.conv2 | (15, 15) | layer3.4.conv1 | (28, 28) |
| layer1.5.conv1 | (10, 10) | layer2.5.conv1 | (15, 15) | layer3.5.conv1 | (28, 28) |
| layer1.5.conv2 | (10, 10) | layer2.5.conv2 | (15, 15) | layer3.5.conv1 | (28, 28) |
| layer1.6.conv1 | (10, 10) | layer2.6.conv1 | (15, 15) | layer3.6.conv1 | (28, 28) |
| layer1.6.conv2 | (10, 10) | layer2.6.conv2 | (15, 15) | layer3.6.conv1 | (28, 28) |
| layer1.7.conv1 | (10, 10) | layer2.7.conv1 | (15, 15) | layer3.7.conv1 | (28, 28) |
| layer1.7.conv2 | (10, 10) | layer2.7.conv2 | (15, 15) | layer3.7.conv1 | (28, 28) |
| layer1.8.conv1 | (10, 10) | layer2.8.conv1 | (15, 15) | layer3.8.conv1 | (28, 28) |
| layer1.8.conv2 | (10, 10) | layer2.8.conv2 | (15, 15) | layer3.8.conv1 | (28, 28) |

Table 24: Layer-wise rank settings of the compressed ResNet-50 model on ImageNet dataset with 2.49× FLOPs reduction. "N/A" denotes we do not compress that layer.

| Layer Name | Rank | Layer Name | Rank | Layer Name | Rank |
|---|---|---|---|---|---|
| FLOPs Reduction = 2.49× | | | | | |
| layer1.0.conv1 | N/A | layer2.3.conv1 | (48) | layer3.5.conv1 | (64) |
| layer1.0.conv2 | (32, 32) | layer2.3.conv2 | (48, 48) | layer3.5.conv2 | (64, 64) |
| layer1.0.conv3 | (32) | layer2.3.conv3 | (48) | layer3.5.conv3 | (72) |
| layer1.1.conv1 | N/A | layer3.0.conv1 | (64) | layer4.0.conv1 | (96) |
| layer1.1.conv2 | (32, 32) | layer3.0.conv2 | (64, 64) | layer4.0.conv2 | (96, 96) |
| layer1.1.conv3 | (32) | layer3.0.conv3 | (72) | layer4.0.conv3 | (96) |
| layer1.2.conv1 | N/A | layer3.1.conv1 | (64) | layer4.1.conv1 | (96) |
| layer1.2.conv2 | (32, 32) | layer3.1.conv2 | (64, 64) | layer4.1.conv2 | (96, 96) |
| layer1.2.conv3 | (32) | layer3.1.conv3 | (72) | layer4.1.conv3 | (96) |
| layer2.0.conv1 | (48) | layer3.2.conv1 | (64) | layer4.2.conv1 | (96) |
| layer2.0.conv2 | (48, 48) | layer3.2.conv2 | (64, 64) | layer4.2.conv2 | (96, 96) |
| layer2.0.conv3 | (48) | layer3.2.conv3 | (72) | layer4.2.conv3 | (96) |
| layer2.1.conv1 | (48) | layer3.3.conv1 | (64) | | |
| layer2.1.conv2 | (48, 48) | layer3.3.conv2 | (64, 64) | | |
| layer2.1.conv3 | (48) | layer3.3.conv3 | (72) | | |
| layer2.2.conv1 | (48) | layer3.4.conv1 | (64) | | |
| layer2.2.conv2 | (48, 48) | layer3.4.conv2 | (64, 64) | | |
| layer2.2.conv3 | (48) | layer3.4.conv3 | (72) | | |

