# OpenReview forum: "ELRT: Towards Efficient Low-Rank Training for Compact Neural Networks"
_ICLR.cc/2023/Conference — Submitted to ICLR 2023_

### Official Review · Reviewer_yVpF · 2022-10-19

**Confidence:** 4
**Correctness:** 3
**Technical Novelty And Significance:** 2
**Empirical Novelty And Significance:** 2
**Recommendation:** 6

**Clarity, Quality, Novelty And Reproducibility:**

Overall the paper is well-organized and motivated.
I think the novelty is very limited. Although the authors provide sufficient analysis to motivate their choices, the core methods are inherited from previous works.

**Strength And Weaknesses:**

Strength:
1) The paper is well-written and easy to follow.
2) Questions in section 4 is well-motivated.
3) It is good to have intensive experiments and latency measured on embedded GPUs.


Weakness:
1. The analysis for the question1 is not convincing. 1) There could be other formulations. For example, what if one perform SVD on each filter, instead of on the reshaped 2D weight matrix? 2) It would be better to show the averaged approximation error across all layers, instead of picking some specific layers.
2. The analysis for question 3 is not novel. The original DSO method is not developed in this work. The authors just reiterate orthogonal regularizations from Bansal et al. (2018).
The two bullet points make Algo. 1 is not novel.
3. No ablation studies on the regularization scaling factor $\lambda_d$.
4. Paper writing: 1) I think the authors used too many underlines, making it hard to find out which part did the authors really want to emphasize. 2) Minor issues: on the 4th row on page five, it should be “lose”, not “loss”

**Summary Of The Paper:**

This work proposes to train low-rank CNNs from scratch. To achieve this, the authors propose Tucker-2 decomposition with low-rankness, and improve the training with orthogonal weight regularizations.

**Summary Of The Review:**

The authors provide sufficient analysis for their design choices and conduct intensive experiments with hardware measurements.
However, as discussed above, my main concern is the core novelty of the method.

---

> ### Author Response · Authors · 2022-11-18
> **Response to Reviewer 4 (2/2)**
>
> #### Q3: The analysis for question 3 is not novel. The original DSO method is not developed in this work. The authors just reiterate orthogonal regularizations from Bansal et al. (2018).
>
> Thanks for the comments. We would like to argue that ELRT has sufficient technical novelty because of four reasons.
>
> First, the prior works focus on full-size training; while ELRT, to the best of our knowledge, is the first work to explicitly and systematically investigate the effect of enforcing orthogonality on low-rank training. The different target training scenario and model formats bring very two different conclusions as follow.
>
> Second, ELRT gives a very different conclusion on the objective that should be enforced with orthogonality. Prior orthogonality-aware full-size training imposes the orthogonality on the entire weight tensors. Instead, our analysis shows that for low-rank training, such regularization should only perform on the factor matrices ($\mathbf{U}^{(1)}$ and $\mathbf{U}^{(2)}$) instead of all the components (including $\boldsymbol{\mathcal{G}}$) that full-size training would do.
>
> Third, ELRT gives a very different conclusion on the suitable regularization format.  “Bansal et al. (2018)” claims SRIP is preferred for full-size training; while for ELRT, after analyzing four regularizations, we conclude DSO is most effective for low-rank training. As shown in Figure 5 and analyzed on page 7 of the paper, this conclusion is derived from our analysis with a perspective of minimizing the energy of residual matrix $\mathbf{R}$. Such discovery and exploration are non-trivial, and it is never explored by “Bansal et al. (2018)”.
>
> Fourth, applying existing regularization to a new problem domain is well-accepted in the deep learning community. As admitted by “Bansal et al. (2018)”, MC and SRIP regularization are essentially proposed in the compressed sensing [R4-1][R4-2]. The key contribution of “Bansal et al. (2018)” is to evaluate the performance of these existing regularization methods in full-size training and then conclude that SRIP is the most suitable one. The novelty of such research effort is acknowledged since it brings new insight to a new research problem. ELRT is the first work to explore the effectiveness of enforcing orthogonality in low-rank training with achieving high model performance.
>
> [R4-1] Decoding by linear programming, Emmanuel Candes
>
> [R4-2] Compressed sensing, David L Donoho
>
> #### Q4: No ablation studies on the regularization scaling factor $\lambda_d$.
>
> Thank you for the valuable comments. Following your suggestion, we conduct the ablation study for training ResNet-20 and ResNet-56 on the CIFAR-10 dataset with different $\lambda_d$. As shown in the following tables and visualize as this [anonymous link](https://anonymous.4open.science/r/Rebuttal-ICLR-E153/lambda.pdf), ELRT training is not very sensitive to  $\lambda_d$ setting. We also add them to the revised manuscript.
>
>
> |  Orthogonality Penalty in ResNet-20 |  FLOPs Reduction 1.98x Accuracy (%) | FLOPs Reduction 2.51x Accuracy (%) |FLOPs Reduction 3.02x  Accuracy (%) |
> |:------------:|:--------------:|:-----------------:|:------:|
> 1e-6   | 91.48  |   91.11    | 90.68
> 5e-4   | 91.64  |   91.16    | 90.80
> 1e-3   | 91.73  |   91.26    | 90.95
> 5e-3   | 91.61  |   91.29    | 90.96
> 1e-2   | 91.66  |   91.22    | 90.88
> 5e-2   | 91.55  |   91.37    | 91.28
> 1      | 91.78  |   91.12    | 91.09
>
>
> |  Orthogonality Penalty in ResNet-56 |  FLOPs Reduction 2.05x   Accuracy (%) | FLOPs Reduction 2.52x Accuracy (%) |FLOPs Reduction 3.01x Accuracy (%) |
> |:------------:|:--------------:|:-----------------:|:-----------------:|
> 1e-6   |  92.82  |   93.24    | 93.28
> 5e-4   |  93.62  |   93.86    | 93.51
> 1e-3   |  93.96  |   94.01    | 93.67
> 5e-3   |  93.95  |   93.87    | 93.64
> 1e-2   |  93.87  |   93.86    | 93.59
> 5e-2   |  94.03  |   93.71    | 93.55
> 1      |  93.87  |   93.75    | 93.64
>
> Q5: Too many underlines and typos.
>
> Thank you very much for pointing this out. Following your suggestion, we have revised the paper to make it more readable and clean. We also correct typos and grammar errors.

---

> > ### Comment · Reviewer_yVpF · 2022-11-26
> > **Thanks for the author's response**
> >
> > I truly appreciate the authors' effort in preparing responses to my questions, esp. results in layer-wise approximation errors and ablation studies on $\lambda$. I would like to raise the score.
> >
> > I encourage the authors to include their arguments on contribution and novelty (abut decomposition and orthogonality) into their paper.

---

> > > ### Author Response · Authors · 2022-11-30
> > > **Thanks for your encouragement**
> > >
> > > Thank you very much for your kind encouragement. Following your suggestion, we will include the analysis and argument on the contribution and novelty of our method to the updated version of the paper. Thank you again.

---

> ### Author Response · Authors · 2022-11-18
> **Response to Reviewer 4 (1/2)**
>
> We sincerely appreciate the reviewer's very constructive comments and suggestions. The following is our response in order of questions and comments raised.
>
> #### Q1: Other possible formulation of matrix decomposition, e.g., SVD perform on each filter?
>
> Thanks for your comments. SVD is a 2-D decomposition method, which cannot be directly applied to each 3-D convolution filter (e.g., 3x3x128). To that end, the filter still needs to be reshaped to 2-D format.
>
> There indeed exists one solution to directly apply SVD to convolution layer without reshaping -- directly apply SVD to each kernel (e.g., 3x3 or 5x5). However, because the kernel size is typically very small with limited full rank value (e.g., 3 or 5), the upper bound of size reduction is very limited. Also, it does not explore the low-rankness exhibited along the input and output channel dimensions.
>
> Overall, we believe high-order tensor decomposition, by its nature, is more suitable to explore low-rankness for 4-D convolution layer than 2-D matrix decomposition. This is because inherently tensor decomposition can extract multi-dimensional correlation in the high-order tensor space.
>
> #### Q2:  It would be better to show the averaged approximation error across all layers, instead of picking some specific layers.
>
> Thank you very much for this valuable comment. The following table shows approximation error across all layer when decomposing ResNet-20 on CIFAR-10 dataset. It is seen that, with the same or even higher compression ratio, high-order tensor decomposition always brings smaller approximation error than the matrix decomposition.
>
> |  Layer Name  |  Weight Shape  | Compression Ratio |  Compression Ratio      | Approximation Error |   Approximation Error     |
> |:------------:|:--------------:|:-----------------:|:------:|:-------------------:|:------:|
> |      -       |        -       |        SVD        | Tucker |         SVD         | Tucker |
> | layer1.conv1 | (16, 16, 3, 3) |        1.59       |  1.59  |        14.16        |  12.71 |
> | layer1.conv2 | (16, 16, 3, 3) |        1.59       |  1.59  |         8.37        |  7.25  |
> | layer2.conv1 | (16, 16, 3, 3) |        1.79       |  1.85  |        25.35        |  17.24 |
> | layer2.conv2 | (16, 16, 3, 3) |        1.79       |  1.85  |         8.49        |  6.80  |
> | layer3.conv1 | (16, 16, 3, 3) |        1.79       |  1.85  |        27.20        |  24.11 |
> | layer3.conv2 | (16, 16, 3, 3) |        1.79       |  1.85  |         8.22        |  7.45  |
> | layer4.conv1 | (16, 32, 3, 3) |        2.00       |  2.11  |        15.40        |  12.46 |
> | layer4.conv2 | (32, 32, 3, 3) |        1.91       |  2.03  |         8.81        |  5.96  |
> | layer5.conv1 | (32, 32, 3, 3) |        1.91       |  2.03  |        15.12        |  12.24 |
> | layer5.conv2 | (32, 32, 3, 3) |        1.91       |  2.03  |         4.13        |  3.20  |
> | layer6.conv1 | (32, 32, 3, 3) |        1.91       |  2.03  |        14.88        |  12.73 |
> | layer6.conv2 | (32, 32, 3, 3) |        1.91       |  2.03  |         3.23        |  2.42  |
> | layer7.conv1 | (32, 64, 3, 3) |        2.01       |  2.01  |        10.52        |  5.90  |
> | layer7.conv2 | (64, 64, 3, 3) |        1.98       |  2.01  |         5.14        |  3.25  |
> | layer8.conv1 | (64, 64, 3, 3) |        1.98       |  2.01  |        12.42        |  7.99  |
> | layer8.conv2 | (64, 64, 3, 3) |        1.98       |  2.01  |         2.70        |  2.21  |
> | layer9.conv1 | (64, 64, 3, 3) |        1.98       |  2.01  |        12.46        |  7.63  |
> | layer9.conv2 | (64, 64, 3, 3) |        1.98       |  2.01  |         0.21        |  0.17  |

---

### Official Review · Reviewer_Afr4 · 2022-10-25

**Confidence:** 4
**Correctness:** 3
**Technical Novelty And Significance:** 2
**Empirical Novelty And Significance:** 2
**Recommendation:** 5

**Clarity, Quality, Novelty And Reproducibility:**

As mentioned in weakness, I think some explanations should be supported by SO regularizer and question 1. I also have a concern about the novelty of a decomposed network from scratch. We can judge the success of a training system by using decomposed layers when there were already full-trained models. But if it is the first time to build NNs w/ new dataset (i.e. not benchmark models/datasets), it is hard to evaluate the performance of networks. It is the fundamental issue on low-precision training or similar approaches. To overcome this question, we need deeper understanding and various evidence on practical networks.

**Strength And Weaknesses:**

Strength
- The proposed method, ELRT, can reduce training cost because the original networks are already decomposed.
- Unlike conventional thinking, they show that low-rank training from scratch can lead to comparable accuracy for image classification models (ResNet/MobileNet).
- They also show accelerated results on various H/W (V100 GPU, FPGA, ASIC, Jetson).

Weakness
- Most experiments are performed on the CIFAR-10 dataset (even including the VGG-16 model). It can show a method is working well, but it doesn’t mean that the method is novel or works well for many NNs for practical applications. Because the key to improved accuracy seems to be adopting a regularizer to the current training system, training redundant models may cause much controversy. To prove the effectiveness of a method, the CIFAR-10 results and ResNet-50(w/ ImageNet) results are not adequate because this model may be not fully regularized.
- There is only one ImageNet result w/ ResNet-50. How about ResNet-18 or other compact networks on ImageNet?
- I think acceleration results are a good way to prove our compression methods are effective and practical. But, how about smaller models including ResNet-18, MobileNet, EfficientNet and so on? Decomposed layers w/ small weights can be not accelerated compared to dense models in highly-parallelized computing systems.
- About Question 1, I have a concern on Figure 3 with MSEs. According to the analysis on MSE, tucker decomposition is selected as a main approach of this paper. However, how can we judge which training system from scratch is better by measuring the MSE of compression? This method is performed w/ decomposed layers and there is no compression step.
- I think there is a lack of explanations and reasons that the orthogonality of U^1 and U^2 can lead to better NNs. But, I’d like to follow other reviewer’s opinions in this aspect.

**Summary Of The Paper:**

While there have been many studies using Low-rank approximation (compression) method for pre-trained models, this paper proposes a new training method from scratch using tucker decomposed models w/o full pre-training steps. Low-rank training from scratch has been conventionally viewed as an insufficient method, which can lead to lower accuracy, but this paper shows comparable accuracy with 2x-3x reduced FLOPs (maybe by adopting SO regularizer).

**Summary Of The Review:**

In my opinion, this paper should strengthen the experimental results and answer some questions on tucker decomposition and regularizer. I understand this method can be applicable to limited networks and dataset, but to prove higher novelty, it should be extended.

---

> ### Author Response · Authors · 2022-11-18
> **Response to Reviewer 3 (2/2)**
>
>
> #### Q4: Why orthogonality of $\mathbf{U}^{(1)}$ and $\mathbf{U}^{(2)}$ can lead to better NNs?
>
> Thanks for the comments. Enforcing orthogonality on a full-size full-rank convolutional layer, with the purpose of improving training performance, has been explored in the literature [R3-6][R3-7]. As analyzed in these existing works, orthogonality in the weights can stabilize the distribution of activations and bring efficient optimizations. Though these prior works focus on full-rank settings, their success demonstrates the potential and benefits of orthogonality for training, motivating our exploration for orthogonality-aware low-rank training.
>
> Moreover, in addition to the benefits revealed in prior full-size training works, we believe enforcing orthogonality is specifically and naturally suitable for low-rank training. From the perspective of linear algebra, it is well known that in principle using an orthogonal-format basis can maximize the capacity of information representation. And that is the reason for 2-D SVD factorization, the decomposed U and V always exhibit self-orthogonality as a unitary matrix with the smallest approximation error. In other words, imposing orthogonality can enhance the capability of preserving information. In the case of matrix reconstruction, this capability leads to smaller reconstruction error; in the case of classification/regression, this capability leads to better learning performance. Similarly, when extending it to high-order Tucker-2 tensor decomposition, the decomposed factor matrices $\mathbf{U}^{(1)}$ and $\mathbf{U}^{(2)}$ also exhibit self-orthogonality after factorizing the full-rank original tensor. Consequently, enforcing orthogonality on the factor matrices in the low-rank training phase, can regularize the update of $\mathbf{U}^{(1)}$ and $\mathbf{U}^{(2)}$ close to the status that can have higher information representation capacity, thereby bringing better performance.
>
> [R3-6] Wang, Jiayun, et al. Orthogonal Convolutional Neural Networks. CVPR 2020.
>
> [R3-7] Bansal, Nitin, et al. Can We Gain More from Orthogonality Regularizations in Training Deep CNNs? NeurIPS 2018.
>
>
> #### Q5: But if it is the first time to build NNs w/ new dataset (i.e. not benchmark models/datasets), it is hard to evaluate the performance of networks. It is the fundamental issue in low-precision training or similar approaches.
>
> Thank you very much for the valuable comments. We believe it is very worthwhile to study low-rank training because of the following reasons.
>
> First, as you mentioned, the potential issue you indicate is not only for low-rank training, but also exists for low-precision training/sparse training from scratch, whose research novelty, to the best of our knowledge, has been well recognized in the community. This is because though these solutions may still be not as mature as the full-size training, the potential benefits in the training/inference efficiency are very promising. To date, many research efforts have been reported on low-precision training/sparse training [R3-8][R3-9]. As a new category in this general "compression-aware training" field, we believe low-rank training should receive sufficient investigation as its counterparts (low-precision training and sparse training) have received.
>
> Second, in practical applications, judging the success of a training system is to see whether if the trained model achieves the target task performance (e.g., classification accuracy) or not. So when it is the first time to build NN with a new dataset, the challenge for evaluating the model performance is the same, no matter whether we plan to train a full-size model or a low-rank model, since we do not have any benchmark/reference model at all. Therefore, training a low-rank model from scratch does not suffer more practical challenging issues than training a full-size model.
>
> Third, hardware vendors have begun to fabricate chips that support training a compact model from scratch. For instance, low-precision training and sparse training have been officially supported by two AI chips from IBM and Moffett AI, respectively [R3-10][R3-11], demonstrating the increasing demand for compression-aware training in real-world business markets. We believe that as the low-rank training solutions become gradually mature, the corresponding hardware support would also occur, just like today's sparse training and low-precision training.
>
> [R3-8] Fu, Yonggan, et al. CPT: Efficient Deep Neural Network Training via Cyclic Precision. ICLR 2021.
>
> [R3-9] Evci, Utku, et al. Rigging the Lottery: Making All Tickets Winners. ICML 2020.
>
> [R3-10] "Meet the IBM Artificial Intelligence Unit.".
>
> [R3-11] Yen, Ian En-Hsu, et al. S4: a High-sparsity, High-performance AI Accelerator.

---

> ### Author Response · Authors · 2022-11-18
> **Response to Reviewer 3 (1/2)**
>
> We sincerely appreciate the reviewer's very constructive comments and suggestions. The following is our response in order of questions and comments raised.
>
> #### Q1: How about ResNet-18 or other compact networks on ImageNet?
>
> Many thanks for pointing this out. Following your suggestion, we report the performance of ELRT on training low-rank ResNet18 on the ImageNet dataset in the following table. It is seen that ELRT shows good performance in training this compact CNN model. We also add it to the revised manuscript.
>
> | Method  | Top-1 Accuracy (%) | Top-5 Accuracy (%) |Inference FLOPs Reduction | Training FLOPs Reduction |
> |--------------|:----------------:|:-----------:|:-----------:|:-----------:|
> |Original ResNet-18 | 69.79 | 89.08 | 1.00x  | 1x |
> |FPGM [R3-1] |68.34 | 88.53 | 1.72x  | <1x|
> |DSA [R3-2]| 68.61 | 88.35 | 1.67x | <1x|
> |SCOP [R3-3]| 68.62 | 88.45 | 1.81x  | <1x|
> |SVDT [R3-4]| N/A | 87.26 | 2.03x  | <1x|
> |TRP [R3-5]| 65.46 | 86.48 |1.81x  | <1x|
> |ELRT (Ours) | 69.80  | 89.29 | 1.40x  | 1.40x|
> |ELRT (Ours) | 68.75  | 88.61 | 1.84x  | 1.84x|
> |ELRT (Ours) | 68.22 | 88.18 | 2.17x  | 2.17x|
>
> [R3-1] He, Yang, et al. "Filter pruning via geometric median for deep convolutional neural networks acceleration". CVPR 2019.
>
> [R3-2] Ning, Xuefei, et al. "Dsa: More efficient budgeted pruning via differentiable sparsity allocation" ECCV 2020.
>
> [R3-3] Tang, Yehui, et al. "Scop: Scientific control for reliable neural network pruning" NeurIPS 2020.
>
> [R3-4] Yang, Huanrui, et al. "Learning low-rank deep neural networks via singular vector orthogonality regularization and singular value sparsification" CVPR worshop 2020.
>
> [R3-5] Xu, Yuhui, et al. "Trp: Trained rank pruning for efficient deep neural networks." IJCAI 2020.
>
>
>
> #### Q2: Acceleration results for smaller models.
>
> Thanks for the comments. We report the practical runtime speedup (per image) for low-rank ResNet-18 on different computing platforms in the following table.
>
> | | FLOPs Reduction  | FPGA Xilinx PYNQZ1 | Desktop GPU Nvidia V100 | Embedded GPU Nvidia Jetson TX2 |
> |-----------------------|:----------------:|:-----------:|:-----------:|:-----------:|
> Original ResNet-18 | 1x  | 85.5ms (1x) | 0.230ms (1x) | 8.17ms (1x)
> ELRT (Ours) | 1.84x  | 50.48ms (1.56x) | 0.212ms (1.08x) | 6.81ms (1.20x)
> ELRT (Ours) | 2.17x  | 48.90ms (1.75x) | 0.203ms (1.17x) | 6.19ms (1.32x)
>
> #### Q3: Why measuring the MSE of compression when there is compression step (Fig. 3)?
>
> Thank you very much for pointing this out. The MSE measurement here is not actually performed in the low-rank training. Instead, we measure the MSE for low-rank compression (given the existence of a pre-trained model) to demonstrate that high-order tensor decomposition (e.g., Tucker-2) brings lower approximation error than 2-D matrix decomposition. Based on such observation, we then decide to adopt Tucker-2 for the low-rank format. In other words, the MSE measurement here is our analysis and exploration to identify the suitable low-rank format when developing the proposed ELRT solution, and it is not actually executed in the training phase at all. We apologize for the caused confusion, and we clarify this in the updated version of the paper.

---

### Official Review · Reviewer_DfQU · 2022-10-27

**Confidence:** 4
**Correctness:** 2
**Technical Novelty And Significance:** 2
**Empirical Novelty And Significance:** 4
**Recommendation:** 5

**Clarity, Quality, Novelty And Reproducibility:**

Clarity: The demonstration and description of the paper are of good clarity.
Quality: The writing and experimental results of the paper are of good quality.
Novelty: The method proposed is of novelty. The importance might be somehow limited given the prior work has demonstrated similar results.
Reproducibility: The authors provided many details about the experimental study. Thus I believe the reproducibility of the paper is good.

**Strength And Weaknesses:**

Strength:
- The paper is well-written. The research direction of speeding up large model training is important.
- The experimental results are thorough and solid.

Weakness:
Overall, my major concern is that the statement "the existing low-rank training solutions are still very limited and do not demonstrate their effectiveness for training modern low-rank CNN models in the large-scale dataset from scratch." is too strong. The papers of [1-3] (which are missing in the reference) have already demonstrated similar promising results. In such a sense, the authors are expected to demonstrate the advantages of ELRT against the low-rank schemes proposed in [1-3]. More weakness is summarized below:

- Using tensor decomposition essentially factories each convolution layer into three "thinner" layers. However, it is well-known that deeper models are harder to train. Thus, does ELRT require any special (if not heavy) hyper-parameter tuning for training such deep low-rank neural networks?
- Given that the transformer network is dominating many application domains currently. I wonder if ELRT also performs well for transformers. Though I understand that the factorization scheme can not be used directly for transformer networks.
-  It was observed by [2] that factorizing the beginning layers can cause accuracy drops. Does ELRT encounter that?
- It seems selecting the "appropriate rank" for different layers is of importance, otherwise, accuracy drops can be observed, e.g., in Table 2. Is there an approach to determine the ranks to select for the network layers for ELRT?


[1] https://arxiv.org/abs/2105.01029
[2] https://proceedings.mlsys.org/paper/2021/hash/84d9ee44e457ddef7f2c4f25dc8fa865-Abstract.html
[3] https://arxiv.org/abs/2006.13347

**Summary Of The Paper:**

This paper proposes ELRT, a low-rank training method that enables training low-rank factorized neural networks from scratch to attain faster training speed and smaller models. Different from the previously proposed low-rank factorization schemes designed specifically for CNNs,  ELRT leverages tensor decomposition. ELRT also utilizes Soft Orthogonal Regularization to further improve the accuracy of the low-rank training. Thorough experimental results on vision tasks are provided to demonstrate the effectiveness of ELRT.

**Summary Of The Review:**

The proposed ELRT method is novel in the sense that it uses tensor decomposition with specialized regularization for training. The method might lack of importance given its performance is very similar to the previously proposed low-rank training schemes (please refer to "Strength And Weaknesses" for more details). I thus urge the authors to include comparisons between ELRT and the aforementioned low-rank training schemes.

---

> ### Author Response · Authors · 2022-11-18
> **Response to Reviewer 2 (2/2)**
>
> #### Q3: Does ELRT require any special (if not heavy) hyper-parameter tuning for training such deep low-rank neural networks?
>
> Thank you for the comments. LERT does not require special hyper-parameter tuning. This is because though ELRT decomposes each convolutional layer to three "thinner" layers, there does not exist a non-linear layer (e.g., ReLU) between these factorized layers, so the overall function of each three "thinner" layer is still a single layer of linear transformation. In other words, the joint effect of $\mathbf{U}^{(1)}$, $\boldsymbol{\mathcal{G}}$, $\mathbf{U}^{(2)}$ in our low-rank model is equivalent to a standard convolutional layer $\boldsymbol{\mathcal{W}} \in \mathbb{R}^{C_{out} \times C_{in} \times H \times W}$. The "effective" depth of the low-rank model is the same as the depth of the standard full-rank model.
>
>
> #### Q4: ELRT for transformers.
>
> Many thanks for the valuable comments. 1) This work mainly focuses on the efficient training of compact CNN models, since to date most pruning, low-rank decomposition, and sparse training works aim to generate compact CNNs. For a fair comparison, ELRT follows the setting of prior works and compares them in Table 1/2. 2) We do believe that imposing orthogonality on transformers should be able to improve performance. This is because from the perspective of linear algebra, the basic component of transformer, e.g., attention head and MLP, is essentially a 2-D matrix, which can be factorized via matrix/tensor decomposition. 3) Due to our limited computing resources, the huge cost of training large-scale transformers, and limited rebuttal time, we will explore applying ELRT for transformers in the future.
>
>
> #### Q5: Does factorizing the beginning layers cause accuracy drops?
>
> Thank you for the comments. As well observed in existing model compression and sparse training works, e.g., [R2-4]-[R2-7], the beginning layers are more sensitive to size reduction. This is because 1) their original sizes are relatively small with limited redundancy; and 2) they are close to raw input data. Therefore, in order to avoid an accuracy drop with the target reduced model size, a common practice in the existing works is to assign more size reduction for the later layers than the beginning layers. Following this convention, when factorizing the entire models, ELRT sets higher ranks for beginning layers and lower ranks for later layers, thereby persevering performance without accuracy drop.
>
> [R2-4] Tang Yehui, et al. "Scop: Scientific control for reliable neural network pruning." NeurIPS, 2020.
>
> [R2-5] Evci, Utku, et al. "Rigging the lottery: Making all tickets winners." ICML, 2020.
>
> [R2-6] Xu, Yuhui, etal. "TRP: Trained Rank Pruning for Efficient Deep Neural Networks", IJCAI 2020.
>
> [R2-7] Lin, Mingbao, et al. "Hrank: Filter pruning using high-rank feature map." CVPR, 2020.
>
> #### Q6: Table 2 shows an accuracy drop for some rank settings. Is there an approach to determine the ranks to select for the network layers for ELRT?
>
> Thank you very much for the valuable comments. The main reason for the accuracy drop observed in Table 2 is the aggressive model size reduction effort instead of the incorrect rank setting. More specifically, because the redundancy of each original model has its limit, when performing too aggressive model size reduction, no matter which type of methods are used, would hurt the model capacity and thus bring an accuracy drop. For instance, accuracy drop is also observed on several pruning works (e.g., SCOP and CHIP) in Table 2 because they adopt high sparsity ratios. As is seen in the table, when the model size reduction is lower, ELRT can even bring higher accuracy than the baseline.
>
> When the target model size/FLOPs reduction budget is given, ELRT can determine the proper ranks in a convenient way without an expensive search. As reported in Table 19-23 in Appendix D, many layers of the models trained by ELRT share the same ranks. Therefore, with the given model size/FLOPs reduction budget, it is easy to calculate and then heuristically set the layer-wise ranks in a manageable way.

---

> > ### Comment · Reviewer_DfQU · 2022-11-26
> > **Thanks for the authors' rebuttal**
> >
> > I would like to commend the authors for their detailed responses to my review. Part of my other concerns are addressed by the author's response, however, my major concern still remains to some extent.
> >
> > By looking at the comparison between ELRT against [R2-1], [R2-2], and [R2-3], it seems the gains are relatively marginal. I fully understand that ELRT allows training smaller low-rank factorized models with higher final accuracy, but how do such parameter reductions further transform to running time speed-ups or memory footprint reductions?
> >
> > Another concern involves that ELRT requires changing the model architecture, which is fine for pre-training-only tasks, e.g., CIFAR-10 and ImageNet training for CNNs. For modern foundation models, does it mean all previously developed efficient fine-tuning and efficient inference methods have to be modified to be compatible with ELRT?

---

> > > ### Author Response · Authors · 2022-12-04
> > > **Thank you for your valuable reply (2/2)**
> > >
> > > Q8: Running time speed-ups or memory footprint reductions?
> > >
> > > Thank you very much for the comments. The following tables show the comparison of the measured runtime speedup between ELRT and [R2-1][R2-2][R2-3] on different hardware platforms. It is seen that ELRT indeed brings faster speed than the prior works. Notice that we approximate some missing rank configurations of [R2-1][R2-2][R2-3] following our rank setting strategy since they did not provide them.
> > >
> > > | Method  | Model | Compression Ratio | Accuracy (%) | Memory Size | FPGA Xilinx PYNQZ1 Runtime | Embedded GPU Nvidia Jetson TX2 Runtime |
> > > |----------------------|:----------------:|:-----------:|:------------------:| :------------------:| :------------------:|:------------------:|
> > > |[R2-1] Low-rank                      | VGGNet              | 10.0x      | 90.71                     | 8.1MB| 2.8ms (6.14x) | 1.27ms (5.16x)|
> > > |[R2-1] Low-rank+FI                     | VGGNet              | 10.0x        | 90.99                     | 8.1MB| 2.8ms (6.14x) | 1.27ms (5.16x)|
> > > |[R2-1] Low-rank+FD                      | VGGNet              | 10.0x                       | 91.57                     | 8.1MB| 2.8ms (6.14x) | 1.27ms (5.16x)|
> > > |[R2-1] Low-rank+SI+FD                      | VGGNet              | 10.0x                            | 91.58                     | 8.1MB |2.8ms (6.14x) | 1.27ms (5.16x)|
> > > |ELRT (Ours)                             | VGGNet              | **12.4x**          | **92.99**             |**4.8MB**       | **2.2ms (7.82x)** | **1.05ms (6.24x)**|
> > >
> > > | Method  | Model | Train Low-rank Model From Scratch | FLOPs Reduction | Accuracy (%) | Memory Size |FPGA Xilinx PYNQZ1 Runtime | Embedded GPU Nvidia Jetson TX2 Runtime |
> > > |----------------------|:----------------:|:----------------:|:-----------:|:------------------:| :----------------:|:-----------:|:-----------:|
> > > [R2-2] FP32                      | ResNet-50   | N           | 1.14x   | 76.43                     | 91.2 MB | 162.2ms (1.06x) | 29.31ms (1.01x)
> > > [R2-2] AMP                      | ResNet-50   | N           | 1.14x                                  | 76.35                     | 91.2 MB |162.2ms (1.06x) | 29.31ms (1.01x)
> > > ELRT (Ours)            | ResNet-50       | Y       | **2.2x**                    | **76.49**                     | **45.6MB** | **85.62ms (2.01x)** | **21.47ms (1.37x)**|
> > >
> > > | Method  | Model | Train Low-rank Model From Scratch | Remaining Parameters | Accuracy (%) | Memory Size | FPGA Xilinx PYNQZ1 Runtime | Embedded GPU Nvidia Jetson TX2 Runtime |
> > > |----------------------|:----------------:|:----------------:|:-----------:|:------------------:| :----------------:|:-----------:|:------------------:|
> > > [R2-3]                       | ResNet-50     | N             | 25.5M                                                                                                   | 75.6                     | 102MB | 172.0ms (1×) | 29.46ms (1×) |
> > > ELRT (Ours)             | ResNet-50       | Y           | **11.4M**                                                                                                 | **76.11**  | **45.6MB**             | **85.62ms (2.01x)** | **21.47ms (1.37x)**|
> > >
> > > Q9:  Since ELRT requires changing the model architecture, does it mean all previously developed efficient fine-tuning and efficient inference methods have to be modified to be compatible with ELRT?
> > >
> > > Thank you very much for these valuable comments! Using ELRT does not need to modify the existing established efficient fine-tuning and inference method because of the following reasons.
> > >
> > > First, as explained in our response to Q3 of Reviewer DfQU, because there does not exist a non-linear layer (e.g., ReLU) between the factorized layers brought by ELRT, the overall function of each three "thinner" layer is essentially still a single layer of linear transformation. In other words, the joint effect of decomposed components $\mathbf{U}^{(1)}, \boldsymbol{\mathcal{G}}, \mathbf{U}^{(2)}$ in our low-rank model is equivalent to a standard convolutional layer $\boldsymbol{\mathcal{W}} \in \mathbb{R}^{C_{out} \times C_{in} \times H \times W}$. From this perspective, ELRT does not significantly "change'' the model architecture. Therefore, the existing used efficient inference and/or training techniques, e.g., Winograd for fast convolution, operator fusion, can still be applied.
> > >
> > > Second, our experiment results also support this conclusion. When applying ELRT, we use the same hyperparameter settings, e.g., learning rate and weight decay, which are adopted in conventional training without modification, demonstrating the generality and suitability of ELRT in different scenarios.

---

> > > ### Author Response · Authors · 2022-12-04
> > > **Thank you for your valuable reply (1/2)**
> > >
> > > Q7: Marginal gain over [R2-1], [R2-2] and [R2-3]?
> > >
> > > Thank you very much for the comments. We would like to argue that, based on the current "evaluation criterion" in the deep learning community, ELRT achieves significant performance gain over those prior works. For instance, [R2-8] (NeurIPS'21) provides 0.02% accuracy increase with 14% extra FLOPs reduction over [R2-9] (ICLR'21) when compressing ResNet-20 on CIFAR-10. [R2-10] (NeurIPS'20) brings 0.06% accuracy increase with 10% extra FLOPs reduction than [R2-11] (ICLR'20) when compressing ResNet-50 on ImageNet. Such gains provided by prior works, even only with less than 0.1% accuracy increase and less than 20% FLOPs reduction, are still widely acknowledged by the community and viewed as a meaningful improvement. This is because as research of efficient deep learning has developed for several years, the existing solutions have already achieved very high performance. It is quite challenging to immediately gain very high improvement (e.g., 5% absolution accuracy increase and/or 2 times FLOPs reduction) over the state-of-the-art works). Instead, the steady improvement, contributed by the entire community in a continuous way, is promoting and will continue to put this research field forward.
> > >
> > > Actually, compared with the above raised two improvements ([R2-8] vs. [R2-9] and [R2-10] vs. [R2-11]), which only has less than 0.1% accuracy increase and less than 20% FLOPs reduction, ELRT shows much better performance improvement. Specifically, ELRT brings at least 1.41% accuracy increase with more FLOPs reduction as compared to [R2-1], 0.06% accuracy increase with 42% extra FLOPs reduction as compared to [R2-2], and 0.51% accuracy increase with 56% extra parameter reduction as compared to [R2-3]. Therefore, based on the current evaluation criterion in the deep learning community, we strongly believe the performance gain provided by ELRT is very meaningful.
> > >
> > > [R2-8] Zhou, Xiao, et al. Efficient Neural Network Training via Forward and Backward Propagation Sparsification. NeurIPS 2021.
> > >
> > > [R2-9] Yuan, Xin, et al. Growing Efficient Deep Networks by Structured Continuous Sparsification. ICLR 2021.
> > >
> > > [R2-10] Tang, Yehui, et al. SCOP: Scientific Control for Reliable Neural Network Pruning. NeurIPS 2020.
> > >
> > > [R2-11] Liebenwein, Lucas, et al. Provable Filter Pruning for Efficient Neural Networks. ICLR 2020.

---

> > > ### Author Response · Authors · 2022-12-11
> > > **More results for Q4: Evaluate ELRT on transformers**
> > >
> > > More results for Q4: Evaluate ELRT on transformers.
> > >
> > > Thank you very much for the valuable comments. We conducted the experiment for the ViT model on the ImageNet dataset. It is seen that ELRT brings significant cost reduction with the similar or even higher accuracy.
> > >
> > > | Method  | Model | FLOPs Remaining | Top-1 Accuracy (%) |
> > > |----------------------|:----------------:|:-----------:|:------------------:|
> > > |[R2-12] Baseline              | ViT-B-16      | 100.0%  |   77.9  |
> > > |ELRT (Ours)            |  ViT-B-16      | **78.8%**  | **78.4**    |
> > > |ELRT (Ours)            |  ViT-B-16      | **59.3%**  | **78.0**    |
> > >
> > > [R2-12] Dosovitskiy, Alexey, et al. An Image is Worth 16x16 Words: Transformers for Image Recognition at Scale. ICLR 2020.

---

> ### Author Response · Authors · 2022-11-18
> **Response to Reviewer 2 (1/2)**
>
> We sincerely appreciate the reviewer's very constructive comments and suggestions. The following is our response in order of questions and comments raised.
>
> #### Q1: The statement "the existing low-rank training solutions are still very limited and do not demonstrate their effectiveness for training modern low-rank CNN models in the large-scale dataset from scratch." is too strong
>
> Thank you very much for your valuable comments. We have revised the manuscript to adjust the tone.
>
> #### Q2: The authors are expected to demonstrate the advantages of ELRT against the low-rank schemes proposed in [R2-1]-[R2-3].
>
> Thank you very much for pointing out the literature. Following your suggestion, we compare ELRT with these related works as follows. We also discuss and compare them in the revised manuscript.
>
> [R2-1] applies spectral initialization (SI) and Frobenius decay (FD) for low-rank training. As shown in the following table, ELRT outperforms different variants of [R2-1] when training VGG on the CIFAR-10 dataset with a higher compression ratio and at least 1.4\% accuracy increase.
>
> | Method  | Model | Compression Ratio | Accuracy (%) |
> |----------------------|:----------------:|:-----------:|:------------------:|
> |[R2-1] Low-rank                      | VGGNet              | 10.0x      | 90.71                     |
> |[R2-1] Low-rank+FI                     | VGGNet              | 10.0x        | 90.99                     |
> |[R2-1] Low-rank+FD                      | VGGNet              | 10.0x                       | 91.57                     |
> |[R2-1] Low-rank+SI+FD                      | VGGNet              | 10.0x                            | 91.58                     |
> |ELRT (Ours)                             | VGGNet              | 12.4x          | 92.99                     |
>
>
> [R2-2] needs a warm-up phase to train a full-rank model in the first epochs, so it does not reduce the overall memory requirement, which is measured by peak memory usage. Also, such a strategy suffers a higher training cost as compared to training a low-rank model from scratch. Instead, ELRT trains a low-rank model from scratch and always keeps the low-rank format during the training. As shown in the following table, ELRT brings higher model accuracy with fewer FLOPs than [R2-2] for training ResNet-50 on ImageNet.
>
>
>
> | Method  | Model | Train Low-rank Model From Scratch | FLOPs Reduction | Accuracy (%) |
> |----------------------|:----------------:|:----------------:|:-----------:|:------------------:|
> [R2-2] FP32                      | ResNet-50   | N           | 1.14x   | 76.43                     |
> [R2-2] AMP                      | ResNet-50   | N           | 1.14x                                  | 76.35                     |
> ELRT (Ours)             | ResNet-50       | Y       | 2.2x                    | 76.49                     |
>
>
> Similar to [R2-2], [R2-3] is initialized with a wide and full-rank model and then decomposes it into a low-rank format after a few epochs. Instead, ELRT performs low-rank training from scratch and always keeps the model stay in the low-rank format during the entire training procedure. As shown in the following table, ELRT shows 0.51\% accuracy increase over [R2-3] with two times model size reduction for training ResNet-50 on the ImageNet dataset.
>
> | Method  | Model | Train Low-rank Model From Scratch | Remaining Parameters | Accuracy (%) |
> |----------------------|:----------------:|:----------------:|:-----------:|:------------------:|
> [R2-3]                       | ResNet-50     | N             | 25.5M                                                                                                 | 75.6                     |
> ELRT (Ours)             | ResNet-50       | Y           | 11.4M                                                                                                 | 76.11                     |
>
>
>
> [R2-1] Initialization and Regularization of Factorized Neural Layers. Khodak, et al.
>
> [R2-2] Pufferfish: Communication-efficient Models At No Extra Cost. Wang, et al.
>
> [R2-3] Principal Component Networks: Parameter Reduction Early in Training. Waleffe, et al.

---

### Official Review · Reviewer_rfLo · 2022-10-31

**Confidence:** 4
**Correctness:** 3
**Technical Novelty And Significance:** 3
**Empirical Novelty And Significance:** 3
**Recommendation:** 6

**Clarity, Quality, Novelty And Reproducibility:**

The paper is good in clarity, quality, and novelty. Also, the reproducibility should not be bad since the authors provide very detailed implementation instructions.

**Strength And Weaknesses:**

Strength

1. This paper provides a very neat and practical solution for low-ranking training with the tucker decomposition. Although the topic seems to overlap with former literature, to the best of my knowledge, the method still holds its novelty.

2. The experiments show promising results, and the actual running times in edge devices convince the effectiveness of the proposed method.

Weaknesses

1. The authors should compare the proposed method with two more families of works if possible: i) the works that employ tensor networks straightforwardly, such as [1, 2]; ii) the works that are trained densely, and compressed with low-rank tensors, such as [3, 4].

[1] Hayashi, K., Yamaguchi, T., Sugawara, Y., & Maeda, S. I. (2019). Exploring unexplored tensor network decompositions for convolutional neural networks. Advances in Neural Information Processing Systems, 32.

[2] Su, Jiahao, et al. "Compact Neural Architecture Designs by Tensor Representations." Frontiers in artificial intelligence 5 (2022).

[3] Lin, Rui, et al. "Hotcake: Higher order tucker articulated kernels for deeper CNN compression." 2020 IEEE 15th International Conference on Solid-State & Integrated Circuit Technology (ICSICT). IEEE, 2020.

[4] Yu, Deli, Peipei Yang, and Cheng-Lin Liu. "Learning-based Tensor Decomposition with Adaptive Rank Penalty for CNNs Compression." 2021 IEEE 4th International Conference on Multimedia Information Processing and Retrieval (MIPR). IEEE, 2021.

2. The structures of CNNs are slightly out-of-date, the authors could try their method on search-based optimal network structures (eg. CoAtNet) or networks with unusual designs (eg. RepLKNet with 31x31 kernels) to enhance the effectiveness of the proposed method.

3. The proposed method could result in matrix multiplications with non-optimal dimensions in practice. In Table 3, the theoretical reduction of FLOPs is somewhat larger than the reduction of running time. Hence the design of the network structure could be difficult and time-consuming.

4. I am still not sure why orthogonality regularization can boost the performance by a large margin since it does not ensure the orthogonality on $\mathbf{U}^{1}, \mathbf{U}^{2}$. If this conclusion still holds under larger and wider CNNs?

**Summary Of The Paper:**

This paper proposes an approach to achieve low-rank training for CNNs. Concretely, the authors consider the form of tucker-2 decomposition to build the convolutional kernel, while during training the orthogonality regularization is imposed on the non-core matrices. The authors experimentally demonstrate the effectiveness of the proposed method using several well-known CNN structures for image classifcation tasks.

**Summary Of The Review:**

This is a solid paper, it could be impactful in both the application of tensor decomposition and network compression and acceleration communities. I vote to accept this paper if the authors could address the aforementioned weaknesses.

---

> ### Author Response · Authors · 2022-11-18
> **Response to Reviewer 1 (2/2)**
>
> #### Q2: The structures of CNNs are slightly out-of-date.
>
> Thanks for the comments. The reason why we evaluate the CNN structures used in ELRT is that most of the existing compression (e.g., pruning and low-rank decomposition) and efficient training (e.g., sparse training [R1-5][R1-6]) report their performance on CNN models such as ResNet50 and so on. To make a fair comparison, we follow the model setting of prior works.
>
> [R1-5] Yuan, Xin, et al. "Growing Efficient Deep Networks by Structured Continuous Sparsification." ICLR 2021.
>
> [R1-6] Zhou, Xiao, et al. "Efficient Neural Network Training via Forward and Backward Propagation Sparsification
> ." NeurIPS 2021.
>
> #### Q3: In Table 3, the theoretical reduction of FLOPs is somewhat larger than the reduction of running time. Hence the design of the network structure could be difficult and time-consuming.
>
> Thanks for your valuable comments. The performance gap between the theoretical FLOPs reduction and practical speedup widely exists for all compact models, e.g., sparse, low-rank, and low-precision formats, not only for the ones trained by ELRT. The main reason is that the runtime latency is jointly determined by several factors in the software/hardware ecosystem, including but not limited to the library, compiler, scheduler, instruction set architecture, and hardware circuit. To date, the computing platform vendors, especially Nvidia, provide very comprehensive optimization for the dense and full-rank model. For instance, CuDNN is highly optimized and customized to accelerate dense GEMM. Such support even occurs at the assembly code and circuit level. On the other hand, the full-stack optimization for compact models, e.g., sparse, low-rank or low precision, is still quite limited as compared to their non-compact counterparts, causing the theoretical benefits of reduced FLOPs can only be partially realized on the hardware.
>
> The network structure of the low-rank model is determined by the ranks. As reported in Tables 19-23 in Appendix D, the layer-wise rank setting in ELRT is very simple. Many layers share the same rank value. Hence it is not a challenging issue to determine a network structure that provides high performance.
>
> #### Q4: Why orthogonality regularization can boost the performance by a large margin since it does not ensure the orthogonality on $\mathbf{U}^{(1)}$ and $\mathbf{U}^{(2)}$.
>
> Many thanks for the valuable comments. Exactly ensuring the orthogonality, i.e., imposing "hard constraint", is computationally expensive since it requires repeated costly SVD during the training phase, eliminating the benefits of low-rank training. On the other hand, enforcing "soft constraint" as the orthogonality regularization, is a low-cost solution to impose the desired orthogonality on $\mathbf{U}^{(1)}$ and $\mathbf{U}^{(2)}$. With proper training, $\mathbf{U}^{(1)}$ and$\mathbf{U}^{(2)}$ can approach ``exact orthogonality", and this near-orthogonality status is already sufficient to preserve most of the important information and enlarge the capacity of information representation, bringing performance improvement.
>
>
> #### Q5: If this conclusion still holds under larger and wider CNNs?
>
> Thanks for the comments. Table 4 in Appendix reports the performance of training WideResNet-28 on CIFAR-10 dataset. We reproduce this table as below. It is seen that ELRT shows high efficiency on the larger and wider CNNs and outperforms the prior works. Notice that here ``$*$" denotes compression ratio since the corresponding works do not report FLOPs reduction.
>
> | Method  | Accuracy (%) | Inference FLOPs Reduction | Training FLOPs Reduction |
> |----------------------|:----------------:|:-----------:|:-----------:|
> |Original WRN-28-8  | 95.60 | 1.00x  | 1x|
> |DST [R1-7] |  94.80 | 10.0x*  | N/A|
> |SFP [R1-8] | 94.22 | 5.00x*  | <1x|
> |DPF [R1-9] | 95.15 | 5.00x*  | N/A|
> |ELRT (Ours) | 95.65 | 5.36x*  | 3.14x|
> |ELRT (Ours) | 95.22 | 6.44x*  | 3.32x|
> |ELRT (Ours) | 94.87 | 11.8x*  | 5.76x|
>
> [R1-7] Liu, Junjie, et al. "Dynamic sparse training: Find efficient sparse network from scratch with trainable masked layers." ICLR 2020.
>
> [R1-8] He, Yang, et al. "Soft filter pruning for accelerating deep convolutional neural networks." IJCAI 2018.
>
> [R1-9] Lin, Tao, et al. "Dynamic model pruning with feedback" ICLR 2020.

---

> > ### Comment · Reviewer_rfLo · 2022-11-22
> > **Thank you for your response.**
> >
> > In the response to R2, the authors claim that the choice of the structures of CNNs is to make a fair comparison.
> >
> > However, in my opinion, fairness and practical usefulness are not conflicted. A method that works on the sota CNNs is much more impactful than a method that only performs better in the "standardized configurations".
> >
> > Therefore, I still suggest the authors show that the method functions well on CNNs such as CoAtNet and RepLKNet.

---

> > > ### Author Response · Authors · 2022-12-11
> > > **Evaluate ELRT on recent models**
> > >
> > > Q6: Evaluate ELRT on recent models.
> > >
> > > Thank you very much for the valuable comments. Following your suggestion, we evaluate ELRT on two latest DNNs, ConvNex [R1-10] and ViT [R1-11] in the following tables. It is seen that ELRT achieves good training performance with considerable cost reduction on both recent CNN and ViT designs. Here we choose these two models instead of CoAtNet and RepLKNet because of two reasons. (1) CoAtNet uses a mixed architecture containing both convolution and attention operations, making it not neither a standard CNN nor a standard transformer. So we believe evaluations on a purely convolution-based CNN (ConvNext) and attention-based transformer (ViT) bring more clear observation and representation. (2) RepLKNet uses a very large-size kernel (31x31), which is not well optimized by Nvidia CUDA. Actually, RepLKNet needs special support from a customized third-party computing engine to achieve practical performance, limiting its generality. Also, RepLKNet adopts a re-parameterization technique, further significantly increasing training time. Therefore, we adopt ConvNext, a very recent CNN (CVPR'22) with similar accuracy to RepLKNet for evaluation.
> > >
> > > | Method  | Model | FLOPs Remaining | Top-1 Accuracy (%) |
> > > |----------------------|:----------------:|:-----------:|:------------------:|
> > > |[R1-10] Baseline              | ConvNext-Tiny      | 100.0%  |   82.1  |
> > > |ELRT (Ours)            |  ConvNext-Tiny      | **76.5%**  | **81.6**    |
> > >
> > > | Method  | Model | FLOPs Remaining | Top-1 Accuracy (%) |
> > > |----------------------|:----------------:|:-----------:|:------------------:|
> > > |[R1-11] Baseline              | ViT-B-16      | 100.0%  |   77.9  |
> > > |ELRT (Ours)            |  ViT-B-16      | **78.8%**  | **78.4**    |
> > > |ELRT (Ours)            |  ViT-B-16      | **59.3%**  | **78.0**    |
> > >
> > > [R1-10] Liu, Zhuang, et al. A ConvNet for the 2020s. CVPR 2022.
> > >
> > > [R1-11] Dosovitskiy, Alexey, et al. An Image is Worth 16x16 Words: Transformers for Image Recognition at Scale. ICLR 2020.

---

> ### Author Response · Authors · 2022-11-18
> **Response to Reviewer 1 (1/2)**
>
> We sincerely appreciate the reviewer's very constructive comments and suggestions. The following is our response in order of questions and comments raised.
>
> #### Q1: The authors should compare the proposed method with two more families of works if possible.
>
> Thank you very much for pointing out the literature ([R1-1]-[R1-4]). Following your suggestion, we compare our proposed ELRT with these prior works as follows. We also discuss and compare them in the revised manuscript.
>
> [R1-1] explores training low-rank CNNs with newly discovered decomposition formats via architecture search. A key drawback is its very high training cost. As reported in [R1-1], it needs 829 GPU days to train ResNet-50 on the CIFAR-10 dataset, limiting the practicality of this approach, especially on the large-scale dataset. Instead, ELRT provides an efficient low-rank training solution with much fewer costs and higher model performance. As shown in the table below, ELRT shows 0.5\% higher accuracy than [R1-1] with two times fewer model parameters for training low-rank ResNet-50 on the CIFAR-10 dataset.
> | Method  | Model | Remaining  Parameters | Accuracy (%) |
> |----------------------|:----------------:|:-----------:|:------------------:|
> | [R1-1]                | ResNet-50           | 22.2M     | 92.2          |
> | ELRT (Ours)           | ResNet-50           | 10.6M     | 92.73         |
>
> Similar to [R1-1], [R1-2] also aims to train low-rank CNNs using new decomposition formats. However, the model obtained by using the method proposed in [R1-2] has limited performance. As shown in the following Table, for training low-rank ResNet-32 on the CIFAR-10 dataset, ELRT shows much higher accuracy (at least 3\%) than [R1-2] with an even higher compression ratio.
> | Method  | Model | Compression  Ratio | Accuracy (%) |
> |----------------------|:----------------:|:-----------:|:------------------:|
> |[R1-2]-mCP              | ResNet-32      | 10.0x  |   82.93  |
> |[R1-2]-mTK             |  ResNet-32      | 10.0x  | 65.75    |
> |[R1-2]-mTT              |   ResNet-32    | 10.0x  | 83.08    |
> |ELRT (Ours)             | ResNet-32     | 10.2x |   87.89   |
> |ELRT (Ours)             | ResNet-32     | 12.7x |  86.60   |
>
> [R1-3] is built on a generalized higher-order Tucker articulated kernel scheme. A key difference between [R1-3] and ELRT is that [R1-3] aims for low-rank compression that requires the existence of a pre-trained model, while ELRT performs low-rank training from scratch without consuming any pre-training cost, significantly reducing training complexity. More importantly, as shown in the following table, ELRT shows better model performance (more than 1\% accuracy increase) than [R1-3] for obtaining low-rank AlexNet on the CIFAR-10 dataset. Here the ranks for Conv2, Conv3, Conv4, and Conv5 are set as [32, 64], [72, 108], [64, 32], and [40, 40], respectively.
> | Method  | Model | Compression  Ratio | Accuracy (%) |
> |----------------------|:----------------:|:-----------:|:------------------:|
> |[R1-3]            |AlexNet     |9.4x    |83.17 |
> |ELRT (Ours)     |AlexNet       |9.4x    |84.45 |
>
> [R1-4] is also a low-rank compression work that requires a pre-training phase. As shown in the following table, even without using any pre-trained high-accuracy model, ELRT still achieves higher accuracy than the pre-trained model-required [R1-4] for obtaining low-rank ResNet-56 on CIFAR-10 and ResNet-50 on ImageNet.
> | Method  | Model | FLOPs  Reduction | Accuracy (%) |
> |----------------------|:----------------:|:-----------:|:------------------:|
> | [R1-4]          | ResNet-56      | 2.8x   | 93.52 |
> | ELRT (Ours)    | ResNet-56   | 3.0x   | 93.67  |
> | [R1-4]       | ResNet-50    | 2.5x   | 75.58  |
> | ELRT (Ours)   | ResNet-50  | 2.5x   | 76.11 |
>
> [R1-1] Hayashi, K., et al. (2019). Exploring unexplored tensor network decompositions for convolutional neural networks. Advances in Neural Information Processing Systems, 32.
>
> [R1-2] Su, Jiahao, et al. "Compact Neural Architecture Designs by Tensor Representations." Frontiers in artificial intelligence 5 (2022).
>
> [R1-3] Lin, Rui, et al. "Hotcake: Higher order tucker articulated kernels for deeper CNN compression." 2020 IEEE 15th International Conference on Solid-State and Integrated Circuit Technology (ICSICT). IEEE, 2020.
>
> [R1-4] Yu, Deli, Peipei Yang, and Cheng-Lin Liu. "Learning-based Tensor Decomposition with Adaptive Rank Penalty for CNNs Compression." 2021 IEEE 4th International Conference on Multimedia Information Processing and Retrieval (MIPR). IEEE, 2021.

---

### Public Comment · ~Andrey_Vinokurov1 · 2023-05-17
**Grouped Convolution issue**

Hello the authors and readers,

Thank you for your research. It looks amazing.

Could you please clarify if you use Tucker Decomposition for Grouped Convolutions?

---

### Decision · Program_Chairs · 2023-01-20

**Decision:**

Reject

**Justification For Why Not Higher Score:**

This paper does not reach the bar of ICLR, which lacks challenge analysis and the authors fail to convince me why these three research questions they address are the fundamental ones to the research question. More details can be found in my meta review.

**Justification For Why Not Lower Score:**

N/A

**Metareview: Summary, Strengths And Weaknesses:**

Based on the collected information from all reviewers and my personal judgment, I can make the recommendation on this paper, **rejection**. Here are the comments that I summarized, which include my opinion and evidence.

**Research Problem**

In this paper, the authors focus on how to train a low-rank CNN model from scratch.

**Related Work**

Several reviewers pointed out that many key missing references.

**Motivation**

Two major motivations of this paper come from the drawbacks of the methods in the same category in terms of performance drop and poor scalability. However, I noticed that the claim is not well supported. For example, Ioannou et al. (2015) and Tai et al. (2015) do not suffer from a performance drop in Appendix A3. And I do not find any evidence of poor scalability, either.

**Presentation**

The authors employ questions to guide the reading experience, which is good. Unfortunately, the logic is not rigorous. I will explain in the following.

**Philosophy**

This paper lacks challenge analysis. In another words, the authors fail to convince me why these three questions are the fundamental ones to the research question. Here I can propose another one that “is that beneficial to add the low-rank constraint on each layer?” In the current presentation, it gives me a feeling that the authors just aim to solve the three questions without a clear motivation. I suggest the authors add the challenge analysis and link it to the three questions.

Moreover, some analyses are not rigorous. Reviewer yVpF pointed out that the analysis for question 1 is not convincing, which I believe the authors agree that in the response. For question 2, several reviewers and I also hold concerns about why orthogonality helps the performance boost. The response that “near-orthogonality status is already sufficient to preserve most of the important information” is too general, which does not convince me.

**Technique**

Technically, the authors employ the DSO regularizer for tucker decomposition.

**Experiments**

(1) $\lambda_d$. Although the authors add the ablation study on $\lambda_d$, I believe $\lambda_d$ is important to the final results (See FLOPs Reduction 2.05x in ResNet-50). The hyperparameter $\lambda_d$ is not clearly illustrated in each setting.

(2) Backbone. The backbone networks used here are relatively small. It would be of high value to conduct the large-scale networks.

(3) Results. The comparisons are incomplete. Some methods mentioned in the introduction and related work parts are not involved for fair comparisons in Table 1. Moreover, the authors report different competitive methods on different backbones.

Feel free to let me know which point is not accurate or whether you want to add some extra points. It is very pleasant to work together with you. Thank you very much.

No objection was raised from the reviewers on the rejection recommendation.

**Summary Of Ac-Reviewer Meeting:**

This is not a borderline paper.